# Polycomb condensates can promote epigenetic marks but are not required for sustained chromatin compaction

Jorine M. Eeftens [1], Manya Kapoor [1], Davide Michieletto[2,3] & Clifford P. Brangwynne [1,4✉]

Organization of the genome into transcriptionally active euchromatin and silenced heterochromatin is essential for eukaryotic cell function. Phase-separation has been implicated in heterochromatin formation, but it is unclear how phase-separated condensates can contribute to stable repression, particularly for heritable epigenetic changes. Polycomb complex PRC1 is key for heterochromatin formation, but the multitude of Polycomb proteins has hindered our understanding of their collective contribution to chromatin repression. Here, we show that PRC1 forms multicomponent condensates through hetero-oligomerization. They preferentially seed at H3K27me3 marks, and subsequently write H2AK119Ub marks. We show that inducing Polycomb phase-separation can cause chromatin compaction, but polycomb condensates are dispensable for maintenance of the compacted state. Our data and simulations are consistent with a model in which the time integral of Polycomb phase-separation is progressively recorded in repressive histone marks, which subsequently drive compaction. These findings link the equilibrium thermodynamics of phase-separation with the fundamentally non-equilibrium concept of epigenetic memory.

[1] Department of Chemical and Biological Engineering, Princeton University, Princeton, NJ, USA. [2] SUPA, School of Physics and Astronomy, University of Edinburgh, Edinburgh, Scotland. [3] MRC Human Genetics Unit, Institute of Genetics and Molecular Medicine, University of Edinburgh, Edinburgh, Scotland. [4] The Howard Hughes Medical Institute, Chevy Chase, MD, USA. ✉email: cbrangwy@princeton.edu

The genome encodes an organism's heritable genetic information, but its differential expression in cells over time enables the variable phenotypic cellular behaviour that underlies biological function. The regulated expression of genes is known to be intimately linked to the structural organization of the genome, which plays a fundamentally important role in gene expression in all eukaryotic cells[1–3]. In general, genomic sequences are packaged into two types of nuclear domains; euchromatin, an "open" state that allows for RNA transcription, and the more compacted heterochromatin, associated with inactive genes. The higher nucleosome density found in heterochromatin is thought to be inaccessible to transcriptional machinery and refractory to chromatin remodelling required for transcription. Compaction is therefore widely accepted as a major hallmark of repressed chromatin, comprised of silenced genes that are not expressed[4,5].

Heterochromatin can be further classified into two types, constitutive and facultative. Constitutive heterochromatin organizes repetitive sequences such as pericentromeric and subtelomeric regions into silent nuclear compartments that are often located near the lamina or around nucleoli. In contrast, facultative heterochromatin consists of transcriptionally silent regions that can become active depending on the context[6]. Constitutive heterochromatin is characterized by trimethylation of histone H3 on lysine 9 (H3K9me3) and the presence of heterochromatin protein 1 (HP1), while facultative heterochromatin is characterized by trimethylation of histone H3 on lysine 27 (H3K27me3)[6,7]. In addition to these distinct histone marks found in the two types of heterochromatin, facultative heterochromatin can be distinguished by the presence of a set of functionally important Polycomb proteins[8,9].

The context-dependent silencing of facultative heterochromatin underscores the importance of understanding how Polycomb proteins facilitate this process. Indeed, while Polycomb proteins are essential for development and cell differentiation and also play a role in X-chromosome inactivation, cell cycle control, and maintenance of repression[10], much about how these multicomponent complexes facilitate compaction and silencing remains unclear. The canonical Polycomb Repressive Complex 1 (PRC1) consists of four core proteins: CBX, PCGF, PHC, and RING, that each have numerous orthologs resulting in different compositions of PRC1[9]. The CBX subunit recognizes methylated H3K27me3, which is deposited there by the related Polycomb Repressive Complex 2 (PRC2). PRC1's RING subunit, the E3 Ubiquitin ligase RNF2, subsequently deposits a second type of histone mark, by ubiquitinating H2AK119[8,11]. PRC1 subunits have been shown to compact nucleosome arrays in vitro[12–14], although the biophysical mechanism driving this compaction is poorly understood.

Recent studies have provided support for the hypothesis that heterochromatin proteins may induce compaction through phase separation. Both HP1 and Polycomb proteins exhibit features characteristic of many proteins that drive phase separation, including oligomerization domains, intrinsically disordered regions, and substrate (chromatin) binding domains[15]. Moreover, both are capable of undergoing liquid–liquid phase separation, which through surface tension forces could directly drive chromatin into a more compact form[16–19]. Phase separation of heterochromatin proteins could also function synergistically with chromatin, as several recent studies have shown that chromatin itself has an intrinsic ability to phase-separate and compartmentalize, in a manner which depends on particular histone marks[20,21]. Phase separation could also potentially explain the mechanism behind heterochromatin spreading, a poorly understood phenomenon in which histone modifications defining the heterochromatin domain expand[22,23]. However, these chromatin marks are typically inherited by daughter cells after division[24–26], while most phase-separated condensates dissolve during mitosis[27]. Indeed, the potential role of liquid–liquid phase separation in the reading and writing of histone marks, and whether phase separation is necessary for chromatin compaction, remains unclear.

Here we show that PRC1 Polycomb subunits can facilitate the formation of multicomponent condensates, which can read and write repressive histone marks. Rather than PRC1 phase separation actively driving compaction, chromatin is instead compacted through the effect of subsequent post-translational modifications, particularly mediated by the ubiquitin ligase activity of RNF2, which is the central node of PRC1 subunit interactions. Thus, while phase separation arises from equilibrium thermodynamic driving forces that do not have memory, we find that it can facilitate non-equilibrium reactions such as epigenetic writing that can endow cells with memory.

## Results

**Canonical PRC1 subunits form condensates.** Polycomb proteins contain oligomerization domains, as well as intrinsically disordered regions (IDRs), and substrate (chromatin) binding domains, characteristic of proteins driving phase separation[28,29] (Fig. 1b, f, j, n). We chose one ortholog of each of the canonical subunits (BMI1/PCGF4, PHC1, CBX2, RNF2/RING1B) and examined their distribution by immunofluorescence (IF). Consistent with prior studies[18,19,30–32], all proteins showed a punctate pattern, including some larger condensate-like clusters (Fig. 1b, f, j, n). To examine how oligomerization could potentially be driving polycomb phase separation, we utilized the recently developed Corelet system[33], a light-activated oligomerization platform (Fig. 1a). Corelets consist of a 24-mer ferritin core, with each ferritin subunit bearing a photo-activatable iLID protein domain. Tagging a protein of interest with iLID's binding partner sspB allows for its reversible, light-dependent oligomerization, which, depending on protein properties, can drive light-activated phase separation. For example, activation of mCh-sspB alone does not lead to formation of condensates (Fig. S1A). Moreover, by varying the concentration ratio of core to sspB, the propensity of any protein to phase separate upon oligomerization can be precisely quantified, by mapping in vivo phase diagrams[33]. First, we checked whether fusion to mCh-sspB altered the endogenous localization of these proteins. BMI1, PHC1, and RNF2 mCh-sspB fusion proteins were expressed < 5-fold higher than the endogenous level, while CBX2 expression was significantly higher, roughly 20-fold (Fig. S1B). All fusion proteins showed similar localization patterns to endogenous PRC condensates (Fig. 1b, f, j, n), with most prominent incorporation into the larger endogenous condensates. Interestingly, in all cases, the nuclear background increased, indicating saturation of the single overexpressed component[28,34].

Fusion to sspB allows us to interrogate the oligomerization-dependent phase behaviour of each of these subunits in cultured human cells. We first examined BMI1, which in addition to its N-terminal RING (DNA binding) domain contains both an oligomerization domain (RAWUL) and IDR[35] (Fig. 1b). Before light activation, BMI1-mCh-sspB was mostly diffuse throughout the nucleus, with a few small areas of higher intensity, consistent with IF and known endogenous localization patterns reported in literature (Fig. 1c)[18,19,30–32]. Upon light activation, many de novo condensates appeared, and the pre-existing puncta grew. The BMI1 condensates recovered rapidly after photo bleaching, indicating a dynamic exchange of the majority of labelled BMI1 molecules within ~2 min, consistent with literature (Fig. 1d and Fig. S1C)[31]. While the BMI1 puncta are close to the diffraction limit, they also appeared relatively round and were frequently observed to undergo liquid-like coalescence with one another (Fig. 1e). PHC1 is another PRC1 subunit with a native

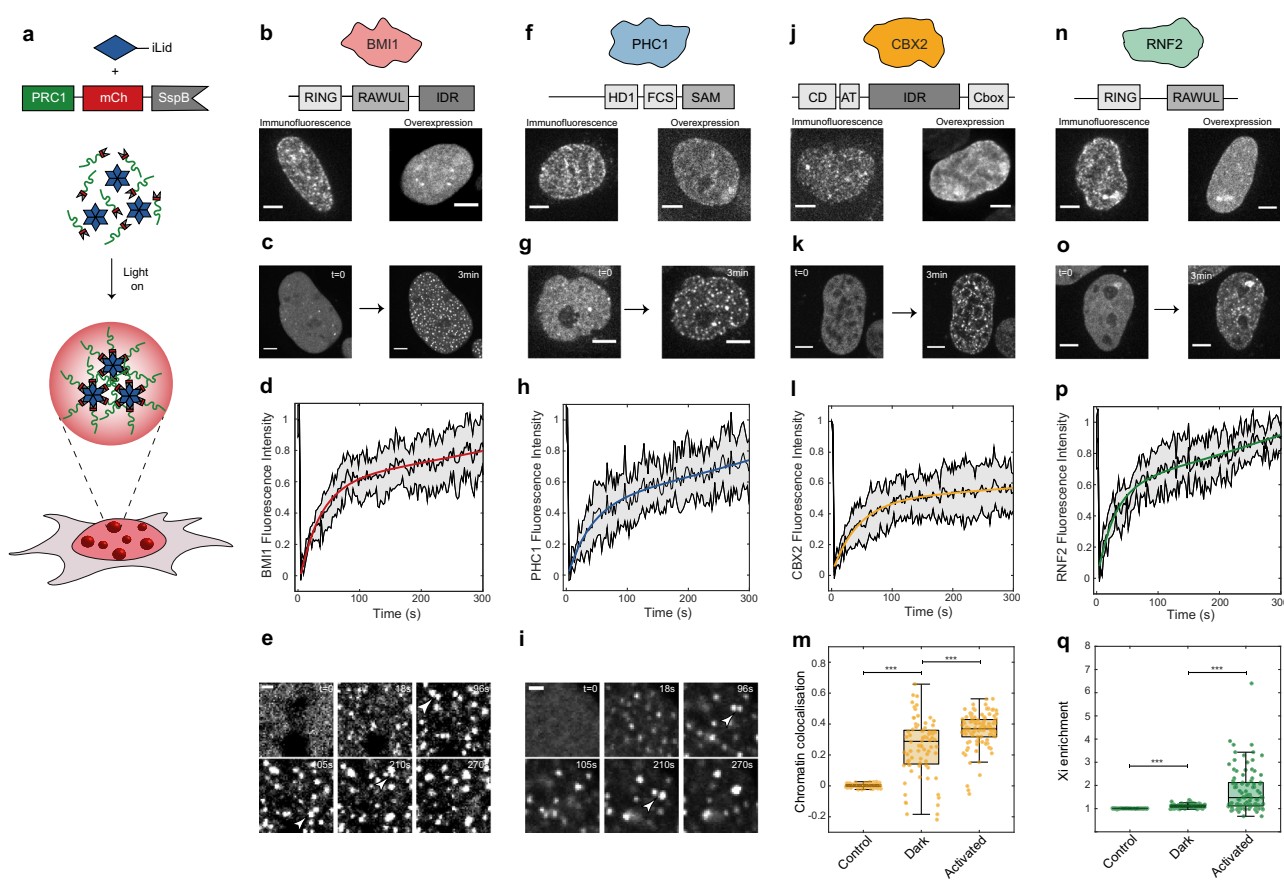

**Fig. 1 PRC1 subunit oligomerization drives phase separation. a** The Corelet system enables light-driven oligomerization to probe protein phase behaviour. **b** BMI1 visualization with immunofluorescence and overexpression. Scale bar is 5 μm. **c** BMI1-mCh-sspB in HEK293 cells with NLS-Ferritin-Corelet before ($t = 0$) and after blue light activation (3 min). Scale bar is 5 μm. **d** FRAP recovery curve of BMI1-mCh-sspB Corelets. Data are presented as mean $+/-$ SD, $n = 5$ cells. **e** BMI1 condensates fuse with one another. Arrows indicate condensates about to fuse. Scale bar is 1 μm. **f** PHC1 visualization with immunofluorescence and overexpression. Scale bar is 5 μm. **g** PHC1-mCh-sspB in HEK293 cells with NLS-Ferritin-Corelet before ($t = 0$) and after blue light activation (3 min). Scale bar is 5 μm. **h** FRAP recovery curve of PHC1-mCh-sspB Corelets. Data are presented as mean $+/-$ SD, $n = 8$ cells. **i** PHC1 condensates fuse with one another. Arrows indicate condensates about to fuse. Scale bar is 1 μm. **j** CBX2 visualization with immunofluorescence and overexpression. Scale bar is 5 μm. **k** CBX2-mCh-sspB in HEK293 cells with NLS-Ferritin-Corelet before ($t = 0$) and after blue light activation (3 min). Scale bar is 5 μm. **l** FRAP recovery curve of CBX2-mCh-sspB Corelets. Data are presented as mean $+/-$ SD, $n = 10$ cells. **m** Pearson correlation coefficient between chromatin (stained with Hoechst, example images in Fig. 4) and mCh-sspB (control, $n = 93$ cells), CBX2-mCh-sspB (in the dark state, $n = 83$ cells), and CBX2-mCh-sspB (after activation, $n = 108$ cells). CBX2-mCh-sspB preferentially localizes with chromatin. This colocalization is increased after activation. Statistical significance, indicated by asterisks, was tested with a two-tailed t test, $P = 3.0190e-24$ for control-Dark and $P = 1.0876e-06$ for Dark-Activated. Central mark of boxplot represents median, bottom and top edges of the box indicate the 25th and 75th percentiles, respectively, and whiskers extend to the most extreme data points not considered outliers. **n** RNF2 visualization with immunofluorescence and overexpression. Scale bar is 5 μm. **o** RNF2-mCh-sspB in HEK293 cells with NLS-Ferritin-Corelet before ($t = 0$) and after blue light activation (3 min). Scale bar is 5 μm. **p** FRAP recovery curve of RNF2-mCh-sspB Corelets. Data are presented as mean $+/-$ SD, $n = 8$ cells. **q** Partition coefficient of mCh-sspB (control, $n = 109$ cells), RNF2-mCh-sspB (in the dark state, $n = 92$ cells), and RNF2-mCh-sspB (after activation, $n = 109$ cells) on the inactive X-chromosome. Statistical significance, indicated by asterisks, was tested with a two-tailed t test, $P = 2.2874e-25$ for control-Dark, $P = 5.5943e-10$ for Dark-Activated. Central mark of boxplot represents median, bottom and top edges of the box indicate the 25th and 75th percentiles, respectively, and whiskers extend to the most extreme data points not considered outliers.

oligomerization domain (sterile alpha motif (SAM)). PHC1 exhibited behaviour similar to that of BMI1: relatively diffuse prior to light activation, with de novo puncta after Corelet activation (Fig. 1g). PHC1 puncta also exhibited nearly complete fluorescence recovery after photobleaching (FRAP) after several minutes and liquid-like fusion behaviour (Fig. 1h, i and Fig. S1D). These data are consistent with BMI1 and PHC1 having an inherent tendency to undergo liquid–liquid phase separation, at sufficient concentration and oligomerization.

CBX2 has previously been shown to undergo concentration-dependent phase separation[18,19]. However, it is unclear whether the same is to be expected in the more complex intracellular environment, particularly given that CBX2 contains a

chromatin-binding chromo domain, and may thus be particularly subject to constraints from the presence of chromatin. Indeed, under Corelet activation, CBX2 displayed behaviour more complex than that of the clearly phase-separated BMI1 and PHC1 condensates. Prior to light activation, CBX2 exhibited a variable intensity across the nucleus, consistent with previously reported endogenous localization[18,19,30] (Fig. 1k). Moreover, instead of forming distinct spherical droplets upon light activation, small chromatin-associated puncta formed, amplifying the pre-activated colocalization pattern (Fig. 1k, m). In particular, CBX2 puncta formed on the heterochromatic regions surrounding the nucleoli (Fig. 1k). These puncta exhibited FRAP recovery of only ~50% after >5 min, indicating that these CBX2

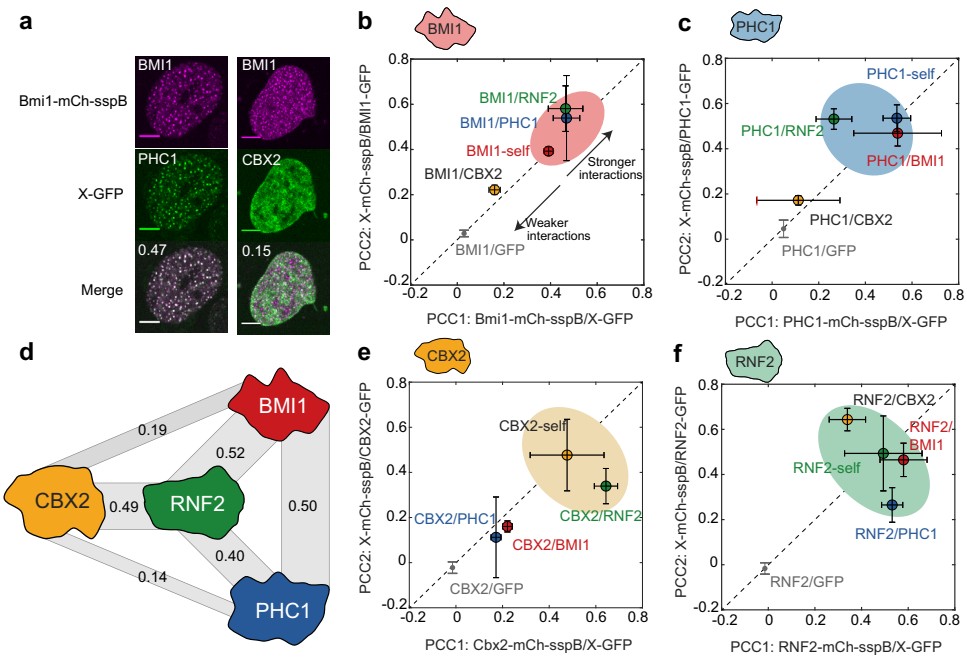

**Fig. 2 Engineered PRC1 condensates recruit endogenous partners. a** Examples of colocalization experiment. BMI1-mCh-sspB Corelets recruit PHC1-GFP, but not CBX2-GFP. Pearson correlation coefficient (PCC) is shown in the upper left corner of the merged image. Scale bar is 5 μm. **b** Recruitment of different PRC1 GFP fusion proteins into BMI1-Corelets (PCC1) (n = 3/7/9/6/5 for BMI1/CBX2/PHC1/RNF2/GFP) and BMI1-GFP into different PRC1-Corelets (PCC2) (n = 3/3/3/6 for BMI1/CBX2/PHC1/RNF2). BMI1 shows self-interaction and significant recruitment of PHC1 and RNF2. Data are presented as mean values +/− SD. **c** Recruitment of different PRC1 GFP fusion proteins into PHC1-Corelets (PCC1) (n = 3/3/6/3/5 for BMI1/CBX2/PHC1/RNF2/GFP) and CBX2-GFP into different PRC1-Corelets (PCC2) (n = 9/4/6/5 for BMI1/CBX2/PHC1/RNF2). PHC1 shows self-interaction and significant recruitment of BMI1 and RNF2. Data are presented as mean values +/− SD. **d** Recruitment data suggest that RNF2 is the central node in the PRC1 complex. Width of grey bars represents relative recruitment strength (average of the two PCC scores). **e** Recruitment of different PRC1 GFP fusion proteins into CBX2-Corelets (PCC1) (n = 3/7/4/10/5 for BMI1/CBX2/PHC1/RNF2/GFP) and PHC1-GFP into different PRC1-Corelets (PCC2) (n = 7/7/3/5 for BMI1/CBX2/PHC1/RNF2). CBX2 shows self-interaction and recruitment of RNF2. Data are presented as mean values +/− SD. **f** Recruitment of different PRC1 GFP fusion proteins into RNF2-Corelets (PCC1) (n = 6/5/5/6/5 for BMI1/CBX2/PHC1/RNF2/GFP) and RNF2-GFP into different PRC1-Corelets (PCC2) (n = 6/10/3/6 for BMI1/CBX2/PHC1/RNF2). RNF2 shows self-interaction and significant recruitment of BMI1, CBX2, and PHC1. Data are presented as mean values +/− SD.

condensates contain a significant immobile fraction, presumably reflecting strong chromatin binding (Fig. 1l and Fig. S1E). RNF2-Corelets exhibited full FRAP recovery, but in this case also colocalizing with the inactive X-chromosome (Xi), with the colocalization enhanced upon light activation (Fig. 1o–q and Fig. S1C–G). This likely reflects the contribution of RNF2 to non-canonical PRC1 in X-chromosome inactivation[36].

Given that these PRC1 subunits exhibit a light-dependent amplification of their endogenous localization patterns and additional de novo puncta formation, we sought to further examine whether native intermolecular interactions are maintained. To test this, we expressed all subunits as green fluorescent protein (GFP)-fusion proteins and assayed their recruitment into PRC1-Corelet condensates (with unlabelled iLID core), quantified by calculating the Pearson correlation coefficient (PCC) between the two fluorescent channels (Fig. 2, see "Methods"). For example, BMI1-Corelet condensates strongly recruit PHC1-GFP, but not CBX2-GFP (Fig. 2a). We found that condensates of each subunit could recruit like proteins, i.e. BMI1-GFP localized into BMI1-Corelet condensates, suggesting that each protein is capable of direct or indirect self-interaction (Fig. 2b, c, e, f and Fig. S1H–K). Moreover, heterotypic interactions were relatively symmetric, e.g. CBX2-Corelets recruited RNF2-GFP, and RNF2-Corelets recruited CBX2-GFP (Fig. 2e, f), while PHC-Corelets do not recruit CBX2-GFP, and CBX2-Corelets do not recruit PHC1-GFP (Fig. 2c, e), underscoring the fidelity of the assay. GFP alone is not recruited to any of the condensates (Fig. 2b–f and Fig. S1H–K). Interestingly, while BMI1, PHC1, and CBX2 exhibit differential

recruitment of the other subunits, RNF2 appears to recruit each with nearly equal apparent strength (Fig. 2f), suggesting that it acts as a central node connecting the three other components, consistent with interaction studies[14,30] (Fig. 2d). We also performed IF imaging of endogenous PRC1 subunits, confirming that our engineered condensates recruit endogenous binding partners (Fig. S1L–O). These interactions are consistent with literature, illustrating the RING–RING interactions between BMI1 and RNF2[37], between BMI1-RAWUL and PHC1-HD1[35], and between RNF2 and CBX2[38]. Thus, the PRC1-Corelet condensates recapitulate key features of endogenous PRC1 complexes and thus represent light-activatable, amplified versions of endogenous PRC1 condensates.

**Hetero-oligomerization contributes to condensate formation.** We next sought to understand the relative contribution of native oligomerization domains, IDRs, and substrate-binding domains to PRC1 condensate formation. (Fig. 3a). We first focused on examining the contribution of the BMI1 IDR to its phase behaviour, by truncating the protein to remove the IDR region (BMI1$^{\Delta IDR}$) (Fig. 3c). Interestingly, BMI1$^{\Delta IDR}$-Corelets showed similar behaviour to the BMI1$^{WT}$. To precisely quantify any subtle difference, we mapped the intracellular phase diagrams for both BMI$^{WT}$ and BMI1$^{\Delta IDR}$. However, the phase diagrams of the two are nearly identical, indicating that the tendency for intracellular phase separation is not driven by the IDR (Fig. 3b, d). Moreover, BMI1$^{\Delta IDR}$-Corelets could still recruit all other PRC1 subunits

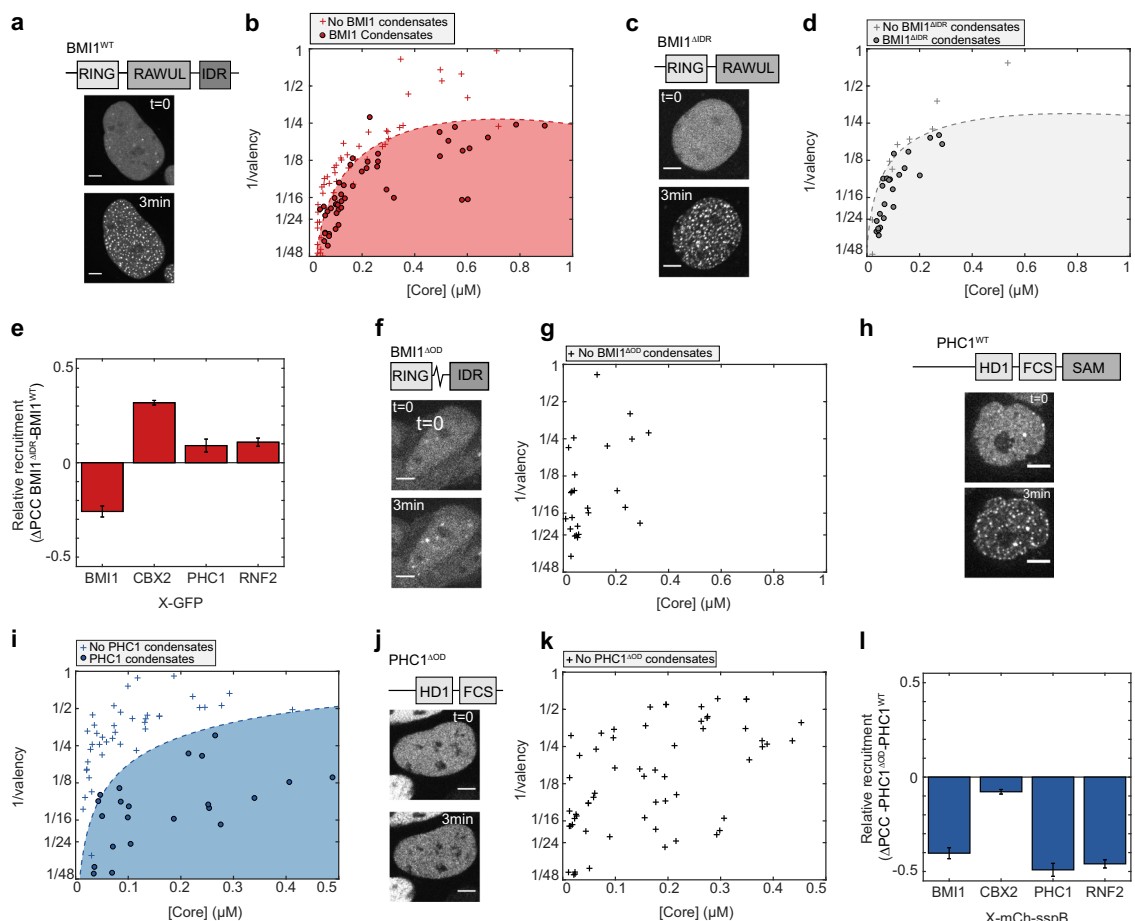

**Fig. 3 Hetero-oligomerization contributes to condensate formation. a** Domain architecture of BMI1$^{WT}$ (top). Images show BMI1$^{WT}$-mCh-sspB in HEK293 cells with NLS-Ferritin-Corelet before ($t = 0$) and after blue light activation (3 min). Scale bar is 5 μm. **b** Phase-diagram of BMI1$^{WT}$. Filled red circles are phase-separated cells, red pluses are non-phase-separated cells. **c** Domain architecture of BMI1$^{ΔIDR}$ (top). Images show BMI1$^{ΔIDR}$-mCh-sspB in HEK293 cells with NLS-Ferritin-Corelet before ($t = 0$) and after blue light activation (3 min). Scale bar is 5 μm. **d** Phase-diagram of BMI1$^{ΔIDR}$. Filled grey circles are phase-separated cells, grey pluses are non-phase-separated cells. **e** Relative change in recruitment of PRC1-GFP fusions for BMI1$^{WT}$ ($n = 3/7/9/6$ for BMI1/CBX2/PHC1/RNF2) and BMI1$^{ΔIDR}$ ($n = 3/3/3/3$ for BMI1/CBX2/PHC1/RNF2). Self-interactions are decreased for BMI1$^{ΔIDR}$ (as indicated by negative ΔPCC), but recruitment of CBX2-GFP is slightly increased (positive ΔPCC). Recruitment of PHC1 and RNF2 shows little change (ΔPCC close to zero). Data represented as difference $+/-$ standard error of the difference. **f** Domain architecture of BMI1$^{ΔOD}$ (top). Images show BMI1$^{ΔOD}$-mCh-sspB in HEK293 cells with NLS-Ferritin-Corelet before ($t = 0$) and after blue light activation (3 min). Scale bar is 5 μm. **g** Phase-diagram of BMI1$^{ΔOD}$, showing that BMI1$^{ΔOD}$ does not phase-separate in this regime. **h** Domain architecture of PHC1$^{WT}$ (top). Images show PHC1$^{WT}$-mCh-sspB in HEK293 cells with NLS-Ferritin-Corelet before ($t = 0$) and after blue light activation (3 min). Scale bar is 5 μm. **i** Phase diagram of PHC1$^{WT}$. Filled blue circles are phase-separated cells and blue pluses are non-phase separated cells. **j** Domain architecture of PHC1$^{ΔOD}$ (top). Images show PHC1$^{ΔOD}$-mCh-sspB in HEK293 cells with NLS-Ferritin-Corelet before ($t = 0$) and after blue light activation (3 min). Scale bar 5 μm. **k** Phase diagram for PHC1$^{ΔOD}$, showing that PHC1$^{ΔOD}$ does not phase-separate in this regime. **l** Relative change in recruitment to PRC1-mCh-sspB fusions for PHC1$^{WT}$ ($n = 9/4/6/5$ for BMI1/CBX2/PHC1/RNF2) and PHC1$^{ΔOD}$ ($n = 3/3/4/5$ for BMI1/CBX2/PHC1/RNF2). Recruitment of all subunits is decreased for PHC1$^{ΔOD}$ (negative ΔPCC). Neither PHC1$^{WT}$ nor PHC1$^{ΔOD}$ recruits CBX2 (ΔPCC near zero). Data represented as difference $+/-$ standard error of the difference.

(Fig. 3e and Fig. S2A), together indicating that the IDR region of BMI1 contributes little to both PRC1 subunit recruitment and Polycomb condensate assembly. Interestingly, however, recruitment of full-length BMI1-GFP was decreased, indicating that BMI1 exhibits weak interactions with itself, likely mediated by homotypic IDR–IDR affinity (Fig. 3e). By contrast, removal of BMI1's RAWUL-oligomerization domain (BMI1$^{ΔOD}$) completely abolished de novo Corelet condensate formation, with only slight growth of pre-existing puncta upon activation (Fig. 3f, g).

It has been previously reported that oligomerization through the SAM domain of PHC1 plays a role in PRC1 clustering[39,40]. First, we quantified the intracellular phase diagram for PHC1$^{WT}$ (Fig. 3h, i). When we truncated PHC1 to remove the SAM-oligomerization domain, PHC1$^{ΔOD}$-Corelets were also unable to

form condensates (Fig. 3j, k). The importance of the SAM domain for phase separation has also been independently shown recently[41]. A comparison of the phase diagram of PHC1$^{ΔOD}$ with PHC1$^{WT}$ shows that PHC1$^{ΔOD}$ is unable to form condensates in the concentration regime that the wild type (WT) does (compare Fig. 3i with k). Thus, oligomerization by Corelets only, without the multiplicative effect of hetero-oligomerization with other PRC1 proteins, is insufficient for PHC1 phase separation. In addition, in contrast to full-length PHC1, PHC1$^{ΔOD}$-GFP was no longer recruited to any of the other PRC1-Corelet condensates (Fig. 3l and Fig. S2B). These data show that hetero-oligomerization domains in BMI1 and PHC1 are essential for multicomponent PRC1 condensate formation.

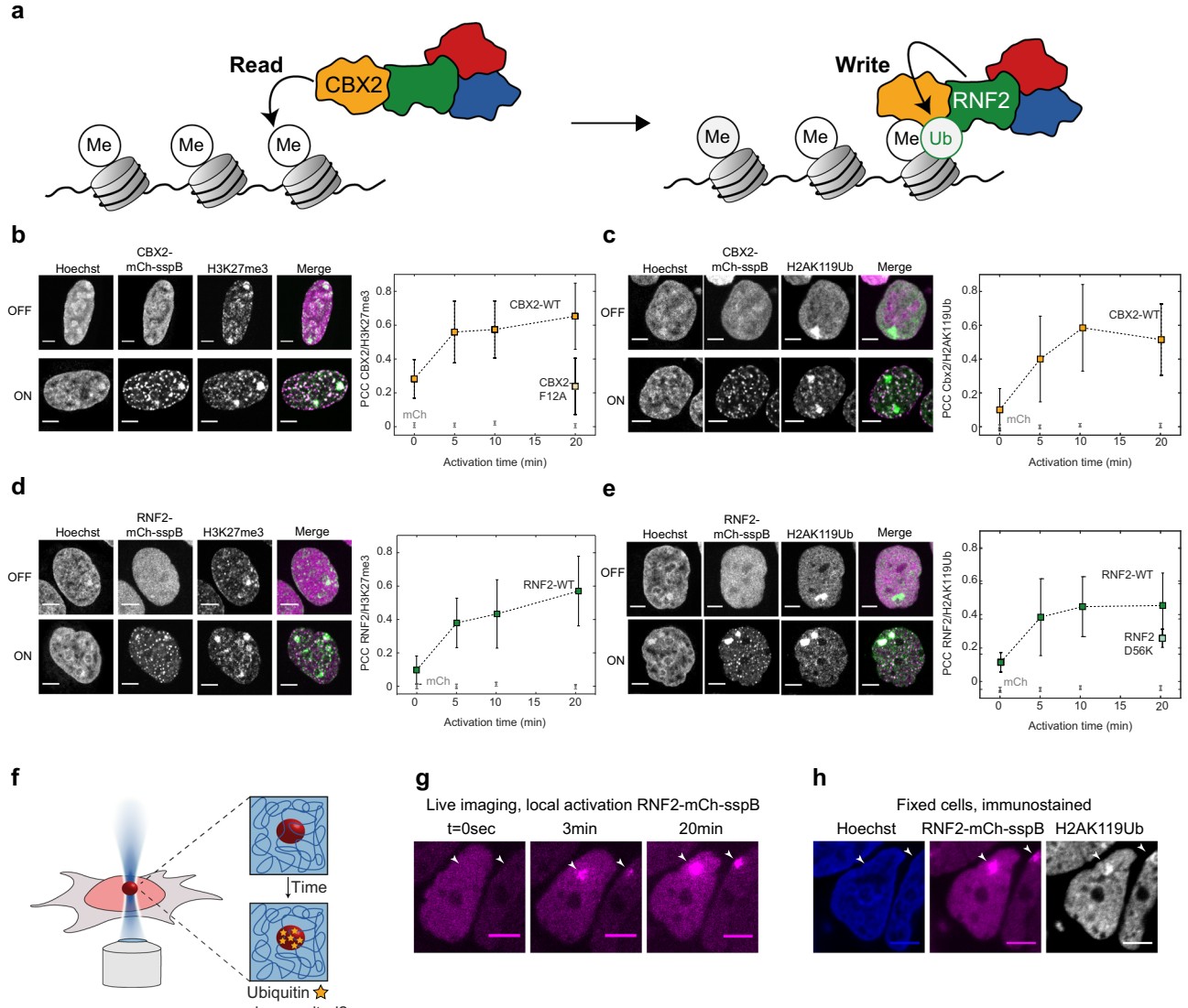

**Fig. 4 PRC1 condensates recognize and write repressive histone marks. a** The Cbx subunit of PRC1 recognizes the H3K27me3 mark, and the RING subunit writes the H2AK119Ub mark. **b** Immunofluorescence on the H3K27me3 mark in fixed CBX2-mCh-sspB Corelet cells, before (OFF) and after (ON) activation. CBX2-mCh-sspB colocalizes with H3K27me3 marks upon light activation ($n = 38/38/36/36$ cells for 0/5/10/20 min for Cbx2, $n = 18/9/7/5$ cells for 0/5/10/20 min for mCh). The CBX2_F12A-mCh-sspB mutant is unable to recognize the H3K27me3 mark ($n = 18$ cells at 20 min). Scale bar is 5 μm. Data are presented as mean values $+/-$ SD. **c** Immunofluorescence staining of the H2AK119Ub mark in fixed CBX2-mCh-sspB Corelet cells, before (OFF) and after (ON) activation. CBX2-mCh-sspB colocalizes with H2AK119Ub marks upon light activation ($n = 40/41/31/37$ cells for 0/5/10/20 min for Cbx2, $n = 14/8/4/14$ cells for 0/5/10/20 min for mCh). Data are presented as mean values $+/-$ SD. **d** Immunofluorescence staining of the H3K27me3 mark in fixed RNF2-mCh-sspB Corelet cells, before (OFF) and after (ON) activation. RNF2-mCh-sspB colocalizes with H3K27me3 marks upon light activation ($n = 28/29/30/25$ cells for 0/5/10/20 min for Cbx2, $n = 18/9/7/5$ cells for 0/5/10/20 min for mCh). Data are presented as mean values $+/-$ SD. **e** Immunofluorescence staining of the H2AK119Ub mark in fixed RNF2-mCh-sspB Corelet cells, before (OFF) and after (ON) activation. RNF2-mCh-sspB colocalizes with H2AK119Ub marks upon light activation ($n = 42/29/29/36$ cells for 0/5/10/20 min for Cbx2, $n = 14/8/4/14$ cells for 0/5/10/20 min for mCh). The RNF2_D56K-mCh-sspB mutant shows less colocalization with the H2AK119Ub mark ($n = 32$ cells at 20 min). Data are presented as mean values $+/-$ SD. **f** Schematic of local activation experiment. Each cell is locally activated in a 1 micron$^2$ area for 20 min, after which the cells are rapidly fixed. Fixed cells are immunostained with anti-H2AK119ub antibody and stained with Hoechst. **g** Local activation of RNF2-mCh-sspB. Arrows indicate region of interest. Scale bar is 5 μm. **h** Fixed cells after local activation of RNF2-mCh-sspB, stained with Hoechst and anti- H2AK119Ub immunofluorescence. Scale bar is 5 μm.

**PRC1 condensates recognize and write repressive histone marks.** CBX2 is known to read H3K27me3 marks, while RNF2 subsequently deposits a Ubiquitin mark (H2AK119Ub; Fig. 4a)[11]. To examine whether this behaviour is recapitulated with our light-activated system, we first probed the interaction of CBX2 condensates with H3K27me3. We fixed cells with CBX2-Corelets before and after light activation and stained for histone marks with IF. In non-activated cells, CBX2 shows a variable

intensity throughout the nucleus, with increased signal on denser chromatin regions (Fig. 4b, "OFF"). The H3K27me3 mark showed a more strongly punctate pattern throughout the nucleus, as well as a bright signal on the inactivated X-chromosomes. Upon light activation, there was no change in the H3K27me3 pattern, and western blots revealed no increase in total H3K27me3 (Fig. S3G). However, light activation causes CBX2 condensates to form in close proximity to these marks, with a

colocalization that increased with activation time (Fig. 4b). Since these cells contain unlabelled, endogenous CBX2, the observed colocalization could potentially reflect CBX2 self-interactions, rather than H3K27me3 binding. To test this, we made a point mutation in CBX2 that is known to prevent it from binding to the H3K27me3 histone mark (CBX2_F12A)[42]. Although this mutated form still appeared to localize to the chromatin (presumably through the AT domain), we found that the overlap between CBX2_F12A condensates and H3K27me3 was reduced to the level of a non-activated cell (Fig. 4b and Fig. S3A, B). This is consistent with CBX2WT condensates colocalizing with H3K27me3 marks due to direct CBX2 reading of these marks. This result also aligns with a recent study which found that H3K27me3 is important for guiding CBX2 to target sites, although CBX2 can also bind chromatin via its AT-hook[43].

We next sought to examine the relationship between the CBX2-recruiting H3K27me3 marks and H2AK119Ub marks. CBX2 is not directly responsible for writing the H2AK119Ub mark, but since CBX2-Corelet condensates strongly recruit the H2AK119Ub writer RNF2 (Fig. 2e and Fig. S1I), we reasoned that these condensates could potentially give rise to H2AK119Ub marks. Before activation, H2AK119Ub is only prominently present on the inactivated X-chromosome (Fig. 4c, "OFF"). However, upon activation and formation of new CBX2 condensates, H2AK119Ub marks start to appear over time, in proximity to these CBX2 condensates (Fig. 4c). Further analysis by western Blot indicates that the total amount of histones modified with H2AK119Ub increases (Fig. S3C); although this increase was not sufficient to establish statistical significance, it is suggestive of CBX2 condensates introducing new repressive marks, rather than redistributing them. We also confirmed that the increase in H2AK119Ub signal is not due to DNA damage induced by the light activation protocol (Fig. S3D)[44]. Thus, our findings strongly suggest that CBX2 condensates are recruited by H3K27me3 marks and can subsequently facilitate the writing of ubiquitination marks onto chromatin within the condensate.

To further examine the causal relationship between H3K27me3 and H2AK119Ub marks, we examined their localization with respect to RNF2-Corelets. Before activation, RNF2 was homogenously distributed, while the H3K27me3 mark showed the familiar punctate pattern, with increased signal on the inactive X-chromosomes (Fig. 4d, "OFF"). Upon light activation, RNF2 condensates began forming at the H3K27me3 marks, similarly to CBX2 (Fig. 4d). Moreover, as with CBX2 condensates, over time H2AK119Ub marks accumulate within RNF2 condensates (Fig. 4e). Consistent with RNF2 directly writing this ubiquitin mark, Corelets formed with an enzymatically dead mutant, RNF2_D56K, showed a slightly reduced colocalization with the H2AK119Ub mark, although the effect was small, likely due to RNF2_D56K retaining an ability to recruit endogenous WT RNF2 and other PRC RING components (Fig. 4e and Fig. S3EF)[45,46]. We did not detect colocalization of BMI1- and PHC1-Corelets with H3K27Me3 or H2AK119Ub (Figure S3H). Taken together, our findings suggest a synergistic epigenetic read–write mechanism by CBX2 and RNF2.

We next sought to determine whether the H2AK119Ub marks could be written at arbitrary genomic locations. A powerful feature of the light-activatable Corelet system is that, by focusing a 488 nm laser on a small subregion of the nucleus, we can locally activate Corelets (Fig. 4f). We activated a single 1-micron$^2$ area in each nucleus for 20 min, fixed the cells, and immunostained for H2AK119Ub (Fig. 4g, h). Remarkably, the H2AK119Ub mark appeared specifically in the locally activated RNF2 condensate (Fig. 4h and Fig. S3I). We also confirmed that these bright H2AK119Ub marks do not simply reflect inadvertent activation on inactive X-chromosomes (Fig. S3I) or as a consequence of localized

illumination (Fig. S3J). Thus, triggering local phase separation of PRC1 condensates drives the writing of repressive histone marks along chromatin segments that are within the condensates.

**PRC1 condensates lead to chromatin compaction.** We next sought to utilize our light-inducible PRC1-Corelets to examine the relationship between phase separation, epigenetic mark reading and writing, and chromatin compaction (Fig. 5a). We investigated whether CBX2-Corelets could compact the DNA by visualizing the chromatin distribution with an miRFP670-tagged Histone2B (H2B-miRFP). Both CBX2-mCh-sspB and H2B-miRFP appear fairly uniform in the dark state (Fig. 5b, $t = 0$ s). Upon activation, CBX2 condensates form, with a variance in the signal that rapidly increases, stabilizing over 5–10 min (Fig. 5c). By contrast, upon activation, the variance in the H2B signal increases in a roughly linear fashion, continuing to increase even after 15 min, when the CBX2 condensates have fully formed (Fig. 5d). Moreover, when CBX2 activation is terminated, the CBX2 condensates rapidly disappear within several minutes (Fig. 5c), but the compaction remains, even after another 20–30 min (Fig. 5d). Thus, PRC condensates rapidly form at H3K27me3-rich chromatin sites, which promotes a slow and steady collapse of chromatin, that can be sustained even after PRC condensates dissolve. Consistent with this picture, the H2B variance appears to be an integral of the CBX2 variance, such that compaction effectively sums over the prior history of PRC1 phase separation (Fig. 5e). This integration over prior PRC phase separation can be further demonstrated by the progressive accumulation of chromatin compaction, over sequential activation cycles (Fig. S4A–C). Fixed cells stained with Hoechst at increasing time points show the same trend, indicating that this is not an artefact of tagged H2B (Fig. S4D, E). The H3K27me3-binding-deficient CBX2_F12A exhibited a markedly different impact on chromatin compaction. While these condensates show the same behaviour as the WT upon activation, with rapidly increasing variance that levels off before deactivation, (Fig. S4K, L), the mutant is unable to compact chromatin, suggesting that interaction with nucleosomes is crucial for inducing compaction (Fig. S4M). Interestingly, however, RNF2-Corelets only exhibit a modest triggering of compaction (Fig. S4F, G). We hypothesize this is due to the limited size of RNF2 condensates. mCh-only, BMI1-, and PHC1-Corelets do not show increased chromatin compaction over time (Fig. S4H–J).

**Repressive histone marks sustain chromatin compaction.** The presence of PRC condensates thus leads to chromatin compaction but is not necessary for maintenance of the compacted state. We hypothesized that this plastic response is due to repressive histone marks, namely ubiquitin, as a consequence of locally concentrated PRC. To examine this possibility, we performed molecular dynamics (MD) simulations of a recolourable polymer model[47]. Briefly, we modelled a chromatin chain as a semiflexible polymer made of beads, with some segments marked as H3K27me3 regions (see "Methods" section for details). Diffusing PRC1 proteins are able to bind these H3K27me3 beads and subsequently write H2AK119Ub on chromatin segments in their proximity. Finally, we assume that ubiquitinated chromatin beads can stick to each other (potentially mediated by other proteins that are not explicitly modelled). After equilibration of the chain, we allow PRC1 to bind H3K27me3 for a certain amount of time and we monitor the average bead density over time (see "Methods"). As proteins bind and modify the histones, the bead density increases steadily, qualitatively similar to our experiments (Fig. 1f, compare Fig. 5d to Fig. 5g). Moreover, when we terminate protein binding (vertical line in Fig. 5f) the chromatin density is sustained. Compaction is not

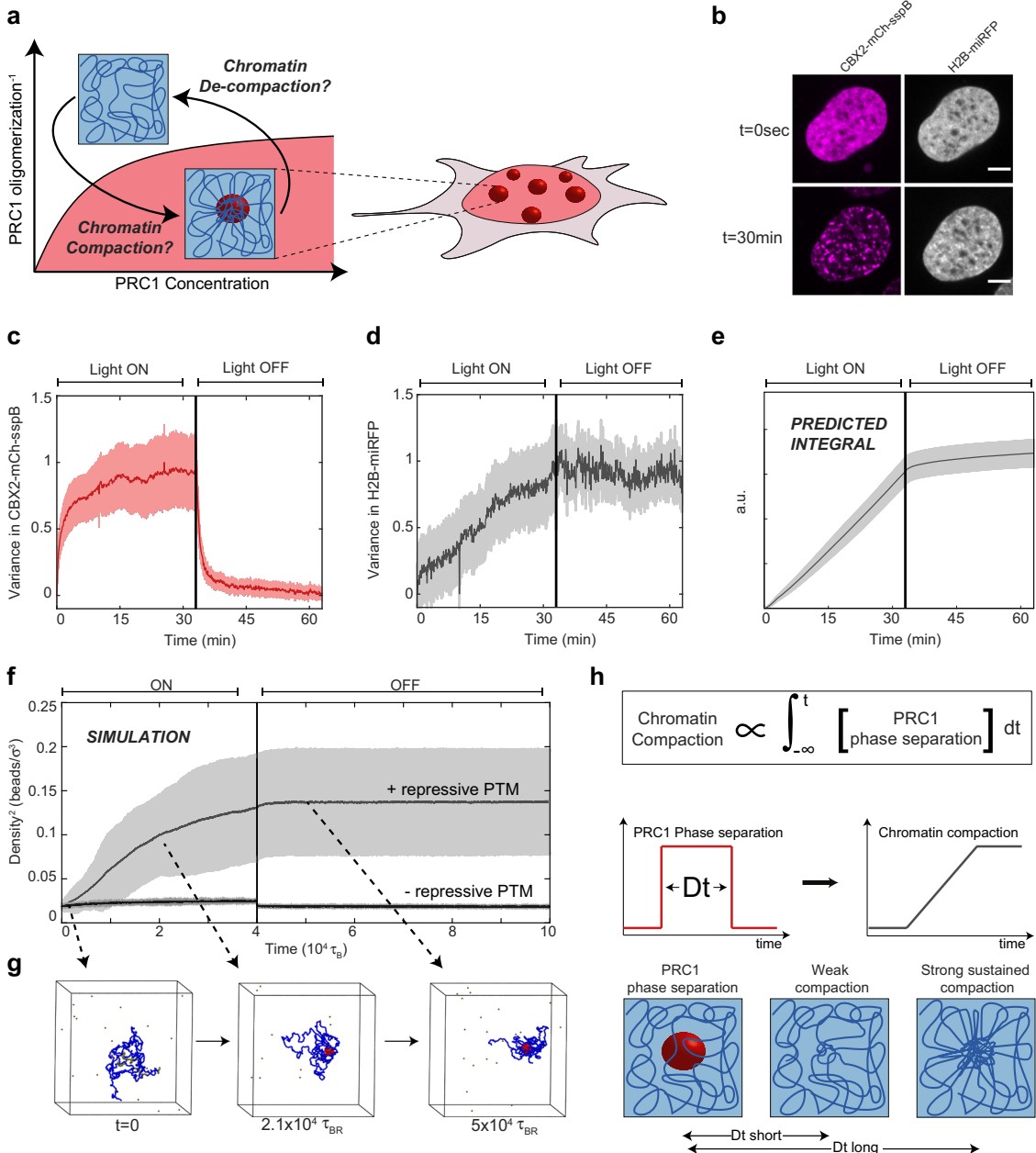

**Fig. 5 PRC1 condensates lead to chromatin compaction. a** Schematic of compaction experiment. After inducing condensate formation, chromatin compaction is examined and tested to determine whether compaction remains after deactivation. **b** CBX2-Corelet cell with H2B-miRFP670 before ($t = 0$) and after activation (30 min). Scale bar is 5 µm. **c** The normalized variance in the CBX2 signal increases rapidly as condensates form and decreases rapidly upon deactivation, as condensates dissolve. Data are presented as mean $+/-$ SD, $n = 10$ cells. **d** The normalized variance in the H2B-miRFP670 signal increases gradually over time, and the high variance remains after deactivation, indicating sustained compaction. Error band represents mean $+/-$ SD. **e** Integral of the CBX2 variance, showing that compaction sums over the prior history of CBX2 phase separation. Error band represents mean $+/-$ SD. **f** Result of molecular dynamics simulation: chromatin bead density over time. Vertical line indicates time point at which proteins no longer bind chromatin. Error band represents mean $+/-$ SD. **g** Snapshot images from molecular dynamics simulation. The chromatin chain in blue, with H3K27me3 patches in yellow, proteins in brown, and H2AK119Ub chromatin beads in red. **h** Long durations of prior PRC1 phase separation can lead to strong sustained chromatin compaction, even in the absence of sustained PRC1 phase separation.

sustained when histones are not modified (Fig. 5f). These simulations illustrate how chromatin modifiers can trigger spreading of silencing marks capable of sustained chromatin compaction, even in the absence of condensates (Fig. 5h)[47].

## Discussion

Here we have used quantitative live cell imaging approaches, complemented by MD simulations, to probe the interplay between phase separation, compaction, and post-translational modifications in the process of heterochromatin formation. We adapted the previously published Corelet system[33] to develop light-activated, multicomponent PRC1 condensates, which allow mapping of subunit recruitment and collective propensity for phase separation. We note that this approach uses light-dependent optogenetic proteins to oligomerize phase separation-prone proteins and may exhibit differences from the biologically regulated oligomerization important for the formation and function of native condensates.

Nonetheless, these synthetic PRC1 condensates recapitulate key features of endogenous PRC1, including the ability to directly recognize and write histone marks associated with facultative heterochromatin. Using the powerful spatiotemporal control of this system, we demonstrate that PRC1 condensation can induce chromatin compaction, but sustained phase separation is dispensable for maintenance of the compacted state.

Our approach enabled interrogation of the role of each of the four core subunits of PRC1, and specific domains of these proteins, in driving phase separation. Strikingly, while BMI1 robustly phase separates into liquid-like condensates upon oligomerization, its IDR did not have a significant impact on condensate formation. This is consistent with recent work showing that, while weak interactions between IDRs can drive phase separation in certain systems[48–53], multicomponent systems in living cells can exhibit more complex behaviour[28,54].

Our light-activated Corelet system confirmed the previously reported interactions between PRC1 components[55,56], while highlighting a heterotypic network of weak affinities among subunits, that drive multicomponent PRC1 condensate formation. This is a more complicated picture than a previously proposed client-scaffold model, in which CBX2 would function as the liquid scaffold, with the rest of the PRC1 components as clients[57]. While each of the PRC1 subunits has an intrinsic capability to form condensates, CBX2 is actually the least dynamic in nature. Instead, CBX2 may act as a localizing "seed" on the chromatin, and through its interactions with the other phase separation-prone PRC1 subunits, particularly BMI1 and PHC1, locally amplifies valence to drive formation of a larger condensate. This local valence is further amplified through oligomerization domains on BMI1 and PHC1, which together with RNF2 appear to serve as valence-amplifying "nodes"[28,39,58,59]. As the catalytic subunit, the E3 ubiquitin ligase RNF2 thus plays a unique role, both as a structural component linking BMI1 and PHC1 to the "reader" CBX2 and also functioning as the "writer," to translate PRC1 condensate formation into localized repressive Ubiquitin marks. Our experiments shed light on the interactions and condensate recruitment of only four canonical PRC1 subunits. Future efforts should be directed towards understanding the full complexity of PRC1 subunit interactions, including non-canonical PRC.

Polycomb condensates have previously been proposed to function by bringing distant chromatin loci together[46,60–62]. This picture distinguishes the catalytic activity (CBX2, RNF2) of PRC1 from its potential role in chromatin remodelling, which could rely on strong oligomerizing of BMI1 and PHC1[39,46,60]. If two genomic loci are bound by a Polycomb condensate, surface tension-mediated pulling of the two loci could bring the loci together, while excluding parts of chromatin not bound by Polycomb proteins[53,63]. Thus, PRC1 condensates could potentially directly remodel the genome, by pulling genomic elements together into dense clusters[64,65].

Despite the attractive simplicity of this PRC1 condensation/compaction picture, our data are inconsistent with PRC1 condensates directly driving chromatin compaction, and instead suggest they indirectly induce compaction by facilitating chromatin marks. Indeed, upon light induction of PRC1-Corelets, chromatin continues steadily increasing in compaction, even after the PRC1 condensates have stabilized, and is sustained even after they have dissolved (Fig. 5d). This behaviour suggests that PRC1 condensates serve as reaction crucibles, a finding that echoes those in a recent study where H2B ubiquitination enzymes were found to form a "reaction-chamber condensate," possibly in a similar manner to PRC1[47]. Consistent with this picture, even after our induced condensate dissolve, the modified chromatin remains in its compacted form, underscoring how compaction cannot be

the result solely of physical forces (e.g. surface tension) imparted by the condensate. Instead, we show that the light-induced condensates trigger the writing of repressive histone marks. These repressive histone marks (both H3K27me3 and H2AK119Ub), rather than the condensates themselves, appear to drive and sustain the compaction of chromatin. Future experiments should be aimed to confirm the direct link between repressive histone marks and compaction.

One possible explanation for how modified chromatin becomes compacted is its intrinsic ability to compartmentalize itself, a possibility that has gained support in several recent studies[20,21]. We envision that, as long as the condensates are present, the dynamic exchange of proteins leads to ubiquitination of the interwoven chromatin, which subsequently compacts itself. Moreover, as the chromatin compacts, new histones move into the condensate and are thus ubiquitinated to further drive compaction. This picture would explain the observed steady increase in compaction even after the condensates' growth saturates. The subsequent ubiquitination would also explain the maintenance of the compacted state after the condensates are dissolved, a result echoed in recent experiments on the role of HP1 in constitutive heterochromatinization[66]. Intriguingly, our findings suggest that phase separation, a reversible process arising from the drive towards thermodynamic equilibrium, can trigger a non-equilibrium change (in particular detailed balance violation through chromatin modifications), with downstream implications for gene expression and cell fate. Our MD simulations of a recolourable polymer model are consistent with this picture and may in the future provide more fully quantitative predictions on the dynamics of epigenetic spreading and its connection to phase separation, a largely underexplored field.

Phase separation in biology has gained considerable attention in recent years, including as a potential mechanism for epigenetic changes. However, as an equilibrium thermodynamic framework, phase separation alone cannot explain heritable epigenetic changes, which must be robustly maintained through replication and cell division. Our results reconcile this key discrepancy by showing how liquid–liquid phase separation can lead to long-lasting non-equilibrium effects, underscoring the complex interplay between the physicochemical driving force of phase separation and the reading and writing of epigenetic information.

## Methods

**Cell culture**. HEK293 cells were used for virus preparation and experiments. Cells were cultured in 10% foetal bovine serum (FBS; Atlanta biological S11150H) Dulbecco's modified Eagle's medium (GIBCO 11-965-118) supplemented with penicillin and streptomycin (Thermo Fisher 15140122) at 37 °C with 5% $CO_2$ in a humidified incubator. On the day before imaging, cultured cells were dissociated with trypsin (Trypsin-EDTA 0.05%, Fisher Scientific 25300054) and transferred on a 96-well glass-bottom dish (Thomas Scientific).

**Plasmid construction**. DNA fragments encoding our proteins of interest were amplified with PCR using oligonucleotides synthesized by IDT (see Supplementary Table S1 for a list of plasmids and Supplementary Table S2 for a list of primers) and CloneAMP HiFi PCR premix. Constructs were cloned using the In-Fusion HD Cloning Kit (Takara Bio 638910). Cloning products were confirmed by sequencing.

**Lentiviral transduction**. All live cell imaging experiments were performed on stably expressing cells, transduced with lentivirus. Lentiviruses were produced by transfecting the desired construct with helper plasmids VSVG and PSP into HEK293T cells with Lipofectamine 3000 (Invitrogen E2311). Virus was collected 2 days after transfection and used to infect WT HEK293 cells in 96-well plates. Three days after addition of virus, cells were passaged for stable maintenance.

**Live cell imaging**. Images were taken on a Nikon A1 laser scanning confocal microscope using a ×60 oil objective (Apo ×60/NA 1.4). The microscope was equipped with a stage incubator to keep cells at 37 °C and 5% $CO_2$. Proteins tagged with Hoechst were imaged using a 405 nm laser, GFP with 488 nm, mCherry with

560, and Alexa647 with 640 nm. Imaging was done on an area of $60 \times 60\ \mu m^2$ (512/512 pixels).

**Statistics and reproducibility**. Sample sizes for live cell microscopy were chosen according to commonly used and accepted standards in the field. For example, for the observations in Figs. 1e, i, 2a, 4g, h, and 5b, three independent biological replicates obtained similar results and technical replicates assured statistical robustness. Representative images were chosen to illustrate the results.

**Corelet activation (global, local, offline)**. Cells were captured in the mCherry channel only, to visualize the sspB component before activation. Cells were then activated with the 488 nm laser (typically for 3 min) with a frame interval of 2 s, while imaging in GFP and mCherry at Nyquist zoom. For local activation, a 1 µm² square is activated. For following compaction over time, the 3 min activation was followed by 30 min of 5 s intervals of activation.

For offline activation, cells were placed on an LED array (Amuza). After activation for an indicated amount of time, 4% formaldehyde is added to the wells. After 10 min incubation, cells are washed with phosphate-buffered saline (PBS; Thermo Fisher 14190250). The cells are then treated with 1:2000 Hoechst (Thermo Fisher H1399) in PBS for 20 min or continued to IF.

**Fluorescence recovery after photobleaching**. Cells were activated globally as described above to acquire steady state. A 1 µm² area in the cluster was then bleached with the 560 nm laser to quench the mCh-sspB component of the condensate. Fluorescence recovery was followed while imaging in both mCh and GFP channels. FRAP experiments were analysed by measuring the mean fluorescence intensity in the bleached area (1 µm²) over time. Error bars represent standard deviation.

**Immunofluorescence**. After activation and fixation as described above, cells were washed with washbuffer (0.35% Triton-X, Thermo Fisher PRH5142, in PBS) for 5 min and permeabilized with 0.5% Triton-X in PBS for 1 h. Cells were then blocked for 1 h using blocking buffer (0.25% Triton-X, 5% FBS, in PBS). Primary antibodies were dissolved in blocking buffer (H2AK119Ub Cell Signal 8240, H3K27me3 Cell Signal 9733, BMI1 Cell Signal 6964, RNF2 Cell Signal 5694: 1:1000, CBX2 Sigma Aldrich HPA023083, PHC1 Sigma Aldrich HPA006973: 1:100) and incubated overnight at 4 °C. The next day, cells were washed 3× for 5 min with washing buffer. The secondary antibody (AlexaFluor 647 goat-anti rabbit Thermo Fisher A-21245, 1:1000) was dissolved in blocking buffer and incubated for 2 h at 4 °C. Cells were washed 3× for 5 min with washbuffer, followed by 20 min incubation of 1:2000 Hoechst. Finally, cells were washed with PBS.

**Western blot**. Adherent cells were offline activated as described above, then washed in ice-cold PBS. Cells were then lysed in RIPA buffer (Thermo Fisher 89900) with benzonase nuclease (Sigma Aldrich E1014-5KU) and scrape harvested to generate whole-cell lysates. Samples were heated to 95 °C for 5 min before running on a 4–12% Bis-Tris gradient gel (Thermo Fisher NP0322) and transferring to a PVDF membrane (Thermo Scientific LC2002). Blots were then blocked in 5% bovine serum albumin (Sigma Aldrich A9418) in TBST (TBS with 0.1% Tween). The blots were then incubated with primary antibody overnight at 4 °C with gentle rocking (H3 Cell Signal 4499, H2AK119Ub Cell Signal 8240, H3K27me3 Cell Signal 9733: 1:1000). The next day, the blots were washed with TBST and incubated with horseradish peroxidase-conjugated secondary antibody (Jackson ImmunoResearch 111-035-144: 1:2000) for 30 min. The blots were then washed with TBST and incubated with ECL (Thermo Fisher 32109) before imaging.

**RNA FISH**. The Stellaris RNA FISH protocol for adherent cells was used for detecting XIST in the cells with RNF2-GFP. The growth medium was decanted; the cells were washed with PBS and then fixed using 4% formaldehyde. After incubation for 10 min at room temperature, the cells were washed twice with PBS and then permeabilized by adding 70% ethanol for a minimum of an hour.

For hybridization, the ethanol was decanted and cells were washed according to the manufacturer's protocol and left to incubate at room temperature for 5 min. Washbuffer was then decanted and the Hybridization Buffer (10% formamide in Stellaris RNA FISH Hybridization Buffer) containing 2% of the Stellaris RNA FISH Probe (Human XIST probe with Quasar 570 Dye) re-dissolved in TE Buffer (10 mM Tris-HCL, 1 mM EDTA, pH 8.0) was added and incubated in the dark at 37 °C for 16 h. The Hybridization Buffer was then aspirated, Washbuffer A was added, and the cells were left to incubate in the dark at 37 °C for 30 min. Finally, the buffer was decanted and the cells were imaged as described.

**Phase diagram construction**. A standard imaging protocol was used on all cells to avoid variability. All activation protocols were 3 min with 2 s intervals. Only cells that were fully in the field of view were considered. Nuclei were manually segmented in ImageJ and the average GFP and mCh fluorescence intensity was determined using the first frame (before activation). Determination of whether or not cells were forming condensates was determined qualitatively by two

independent observers, one of whom was blinded to experimental conditions. The assessments of the two observers were consistent in nearly all cases. The few cells on which observers disagreed were excluded from the results. FCS calibration curves were used to determine the mCh and GFP concentrations from the fluorescence intensities as described in Bracha et al.[33]. Valence was measured as the ratio of sspB-fused protein to core.

**Partition coefficient, PCC, and variance determination**. To determine the partition coefficient on the X-chromosome, nuclei and X-chromosomes were segmented using Matlab, and the fluorescence intensity on the X-chromosome and elsewhere in the nucleus were determined. A partition coefficient of 1 indicates that there is an equal amount of fluorescent protein localizing on the X-chromosome and elsewhere in the nucleus; a partition coefficient of 2 indicates that there is twice as much localization on the X-chromosome as elsewhere in the nucleus.

To estimate the degree of colocalization between two channels, first, the nucleus was segmented with Matlab. As they can heavily skew the degree of colocalization, the nucleoli and inactivated X-chromosome were excluded. Then the pixel-by-pixel correlation was determined using PCC defined as:

$$PCC = \frac{\sum_{i=1}^{n}(x_i - \bar{x})(y_i - \bar{y})}{\sqrt{\sum_{i=1}^{n}(x_i - \bar{x})^2}\sqrt{\sum_{i=1}^{n}(y_i - \bar{y})^2}} \quad (1)$$

where $n$ is the sample size (number of pixels) and $i$ is the individual pixel index, with $x_i$ and $y_i$ the value in that pixel for the two channels and $\bar{x}$ and $\bar{y}$ the sample means.

In the case of co-expressed PRC proteins (shown in Fig. 2), the analysis was done in a similar manner, with the exception that the inactivated X-chromosome was not excluded from analysis. The symmetry of the interactions can be seen from the fact that all data points lie relatively close to the diagonal.

For variance determination, the nucleus was segmented in every frame, with nucleoli excluded. For each time point, the variance in pixel intensities was determined in each channel as:

$$V = \frac{1}{n-1}\sum_{i=1}^{n}|A_i - \mu|^2 \quad (2)$$

where $n$ is the sample size (number of pixels), and $i$ is the individual pixel index, with $A_i$ the value in that pixel and $\mu$ the mean. The variance over time was normalized from 0 to 1 per trace, and we plot the result averaged over the number of cells specified in the figure caption.

**Computational details**. We have modelled a region of chromatin as a semiflexible polymer made of $N = 1000$ beads of size $\sigma = 30$ nm = 3 kbp and with persistence length $l_p = 3\sigma = 90$ nm[67]. The excluded volume interactions between chromatin beads (including consecutive ones along the DNA) obey the shifted and truncated Lennard–Jones (LJ) potential:

$$U_{LJ}(r, \sigma) = 4\epsilon\left[\left(\frac{\sigma}{r}\right)^{12} - \left(\frac{\sigma}{r}\right)^{6} + \frac{1}{4}\right] \text{for } r \leq r_c \quad (3)$$

and 0 otherwise. In this equation, $r$ denotes the separation between the bead centres. The cutoff distance $r_c = 2^{1/6}\sigma$ is chosen such that only the repulsive part of the LJ is used for DNA beads, i.e. there is no attraction. Consecutive monomers along the DNA contour length are connected by the finitely extensible nonlinear elastic (FENE) potential, given by

$$U_{FENE}(r) = -0.5\,k\,R_0^2\log\left(1 - \left(\frac{r}{R_0}\right)^2\right) \text{for } r \leq R_0 \quad (4)$$

and infinity otherwise. In this equation $k = 30\epsilon/\sigma^2$ is the spring constant and $R_0 = 1.5\sigma$ is the maximum extension of the FENE bond. The persistence length for the chromatin is introduced by adding a bending energy penalty between triplets of consecutive beads along the DNA as

$$U_{bend}(\theta) = k_\theta(1 + \cos\theta) \quad (5)$$

where $\theta$ is the angle formed between adjacent bonds, i.e. $\cos^{-1}\left(t_i t_{i+1}/|t_i||t_{i+1}|\right)$ with $t_i$ the tangent at segment $i$, and $k_\theta = 3\,k_B T$ the bending constant.

The chromatin beads are "coloured" in order to account for their epigenetic state, which in turn encodes for different interactions with proteins[67]. At the start of the simulation we set 85 beads—chosen in between the 400 and 600 beads in the chromatin region (for all our simulation replicas)—to represent segments marked as H3K27me3. The rest of the polymer is set to a neutral colour.

The protein bridges (CBX2/RNF2) are represented by 20 spherical beads of size $\sigma_b = 30$ nm (1 protein per 150 kbp of DNA). They can bind regions of the chromatin that are labelled by H3K27me3 marks. The binding is modelled by a LJ potential as described above, where the cutoff is now set to $r_c = 1.8\sigma$ to account for an attractive region and the strength of the interaction is set to $4\,k_B T$.

Every 10 Brownian times (a Brownian time $\tau_B = \gamma\sigma_b^2/k_B T$ is the time it takes for a bead to diffuse its own size, where $\gamma$ is the friction coefficient) we attempt a recolouring of all the chromatin beads. A recolouring attempt proceeds as follows.

A random bead along the chromatin is selected. If a CBX2/RNF2 protein is within 60 nm from that bead and it has a H3K27me3 mark, then the selected bead

becomes ubiquitinated. At this stage, we distinguish between the two following scenarios: beads that were neutral become only ubiquitinated, whereas beads that harboured the H3K27me3 can bear both K27me3 and Ub marks. Finally, the last assumption of our model is that regions of the chromatin that have the Ub mark stick to each other. This attraction is again modelled as a LJ potential with $r_c = 1.8\sigma$ as cutoff and interaction strength set to $3\,k_B T$.

**MD simulations**. The static and kinetic properties of the systems are studied using fixed-volume and constant-temperature (NVT) MD simulations with implicit solvent and periodic boundary conditions. MD simulations are performed using the LAMMPS package (http://lammps.sandia.gov/). The equations of motion are integrated using a velocity-Verlet algorithm, in which all beads are weakly coupled to a Langevin heat bath with a local damping constant set to $\xi = 1/\gamma = 1$ and mass $m = 1$ so that the inertial time ($m/\gamma$) equals the LJ time $\tau_{MD} = \sigma\sqrt{m/\epsilon} = 1$ and also the Brownian time[67]. The integration time step is set to $\triangle t = 0.01\,\tau_{MD}$.

To monitor the collapse, we compute the evolution of the radius of gyration defined as

$$R_G = \frac{1}{N}\sum_{i=1}^{N}\left(r_i - r_{CM}\right)^2 \qquad (6)$$

where $N$ is the number of beads in the polymer and $r_{CM}$ is the position of its centre of mass. We also compute the local monomer density measured as the average number of beads within a sphere of radius 60 nm centred in each of the chromatin beads and divided by the volume of a sphere of radius 60 nm. The plots in the main text show the evolution of the local density squared for comparison to the experimental metric variance and are averaged over 36 independent replicas. In each of the replicas, the chromatin is initialized as a random walk and allowed to equilibrate for $10^6\,\tau_B$.

**Reporting summary**. Further information on research design is available in the Nature Research Reporting Summary linked to this article.

## Data availability

The data that support this study are available from the corresponding author upon reasonable request. Raw microscopic datasets are available upon request. Source data are provided with this paper.

## Code availability

Code used for this manuscript is available at https://git.ecdf.ed.ac.uk/dmichiel/prc_ub_recoloring.

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

## Acknowledgements

We thank Mike Levine, Tom Muir, as well as members of the C.P.B. laboratory for discussions and comments on the manuscript, including Dan Bracha, Josh Riback, and Amy Strom. This work was supported by the Howard Hughes Medical Institute and grants from the NIH 4D Nucleome Program (U01 DA040601). J.M.E. is supported by a Rubicon Grant (NWO). D.M. is a Royal Society University Research Fellow and is supported by the ERC (TAP, 947918).

## Author contributions

Conceptualization: J.M.E. and C.P.B. Analysis software: J.M.E. Formal analysis: J.M.E. Investigation: J.M.E. and M.K. Simulations: D.M. Writing—original draft: J.M.E. and C.P.B. Writing—review and editing: all authors. Supervision: C.P.B.

## Competing interests

C.P.B. is a co-founder and consultant for Nereid Therapeutics. The other authors declare no competing interests.
