## [Peer Review File · Nature Communications]

REVIEWER COMMENTS

Reviewer #1 (Remarks to the Author):

The present work “Epigenetic marks as a time integral over prior history of Polycomb phase separation” reports that Polycomb Repressive Complex 1 (PRC1) condensates drive chromatin compaction but dispensable for maintaining this compaction. The authors sequentially demonstrated or observed: 1) Core subunits of PRC1 can phase separate in an optogenetic corelet system. 2) PRC1-subunit condensates can recruit other core subunits. 3) Oligomerization domains are responsible for condensate formation. 4) Cbx corelet condensates are localized in H3K27me3 by direct reading, and drive H2AK119Ub writing by recruiting RNF2 proteins into condensates. In addition, RNF2 corelet condensates recruit Cbx subunits, are localized around H3K27me3 and progress H2AK119Ub writing. 5) Cbx corelet condensates drive chromatin compaction but dispensable for maintaining chromatin compaction. These observations provide important findings regarding the relations between Polycomb phase separation, histone modification, and finally chromatin compaction.

There are, however, several concerns in this manuscript. For example, while it is stated that Polycomb phase separation can induce chromatin compaction but not for maintenance of the compacted state, RNF2 condensates showed only a slight compaction. Therefore, I recommend addressing following issues before considering publication of this work in Nature Communications.

1. In Figure 3, the idea of ‘hetero-oligomerization’ is not clear. What are the components for hetero-oligomerization? (ferritin & oligomerizing domain? Or multiple oligomerizing domains?). If it is hetero-oligomerization of Corelet & PRC1 ODs, then it must be carefully discussed when using observed data for explaining chromatin compaction, since this is not really natural conditions in cells.
2. Relative co-localization is really important to explain many data, particularly for those in Figure 4 and related supporting figures (e.g. Figure 4B. Figure S3A,B). Overlap images will be useful, or can we have more quantitative data on these co-localization data?
3. Considering the error bars, I cannot find a clear H2AK119Ub signal increase by light ON in the western blot data (Figure S3C). The PCC data of Figure 4C, however, show clear increases by light activation. Based on these data, it looks like that H2AK119Ub signals are re-distributed by Cbx2-Corelets rather than increased. This issue must be clarified.

4. Since many data contain both cell imaging with fused fluorescent proteins (expressed proteins only) and IF imaging with antibodies (expressed proteins and endogenous proteins), it is very important to understand relative levels of expressed proteins and endogenous proteins. In addition, protein expression levels often largely vary cell-to-cell. Please provide these data and discuss the effects of endogenous proteins in all data explanation (although the author did explain the effects of endogenous proteins for some data).

For example, to better understand Figure S3E (RNF2_D56K data), where this mutation did not affect Ub patterns much, relative levels of endogenous RNF2 will be important. Ideally, one would like to directly engineer the endogenous RNF2 (and others) gene to fuse various proteins.

5. Figure S4FG, where 'RNF2-Corelets only exhibit a modest triggering of compaction' cannot be explained by the present model. RNF2-Corelets behave (processes like H3K27Me as well as H2AK119Ub) just like Cbx2-Corelets as shown in Figure 4. In addition, RNF2 can recruit other PRC1 components most effectively as shown in Figure 2. Based on these data, RNF2-Corelets must also trigger chromatin compaction as efficient as Cbx2-Corelets. The authors hypothesize this is due to the limited size of RNF2 condensates. However, there is no experimental evidence for this statement. Maybe, condensate sizes could be varied by controlling corelet valency or protein expression levels to examine this statement.

In addition, corelet condensates of other two proteins, Bmi1 & PHC1, among PRC1 components must also be examined for their influence on chromatin compaction to support the present model.

6. The variance determination procedure was only briefly described. Please provide a more detail method for variance determination in Figure 5 and Figure S4.

Reviewer #2 (Remarks to the Author):

In this paper, Eeftens and colleagues use cell imaging and optogenetics to analyze the composition of condensates induced by clustering of PRC1 subunits (through the recently developed Corelet system) and the effect of condensates on H2A119ub and chromatin organization. This implementation of the Corelet system is the first report of targeted optogenetic manipulation of PRC1 condensate formation.

From the results of these experiments, they draw three main conclusions: 1) condensate formation is driven by hetero-oligomerization; 2) condensates can "read" and "write" histone modifications

(H3K27me3 and H2A119ub, respectively); 3) chromatin compaction is not a direct result of phase separation but a secondary result of histone modifications so that it is not required to maintain chromatin compaction. The authors suggest an interesting model in which the persistence of chromatin compaction/histone modification after dissolution of condensates could allow the effects of cycles of condensate formation to be integrated over time to maintain a repressive chromatin state.

The authors have applied an innovative method for inducing condensate formation, which is especially powerful for studying the dynamics of condensates and how they relate to chromatin organization. While the low resolution of the methods used to analyze chromatin effects make it difficult to connect these findings to gene-specific regulation by PRC1, I think the findings and perspective will be of considerable interest to the field. However, I have several major conceptual and technical concerns with respect to the key conclusions. I think that some additional experiments, additional controls, and additional quantification are required, as is additional information about the details of the experiments. The new results should also be interpreted more precisely with respect to published work on PRC1 (particularly work on Cbx2 from the Ren and Kingston groups, and on Phc2 from Koseki).

Major concerns:

1) The authors conclude that hetero-oligomerization drives formation of condensates but this is not clearly supported by their data. The literature suggests an alternative model, namely that both CBX2 and PHC1 can form condensates in cells that do not depend on PRC1 assembly (but can include PRC1 subunits). None of the experiments provided in this manuscript refute this interpretation.

Previous work on both Cbx2 (Ren and Kingston labs) and Phc2 (Koseki lab) has demonstrated that both of these proteins can form clusters/condensates in cells without interacting with other PRC1 subunits. For Cbx2, this was shown by analyzing condensates in cells lacking Ring1B/Ring1A (no PRC1 formed) as well as those lacking Bmi1/Mel18 (no canonical PRC1 formed). For Phc1 (Isono et al., *Dev. Cell*, 2013 doi: 10.1016/j.devcel.2013.08.016.), clustering depends on the polymerization interfaces of the SAM, but not on the HD1 domain that mediates assembly into PRC1. These strategies to test the role of interactions with other PRC1 subunits should be applied in the Corelet system:

a) For CBX2, deletion of the cbox domain should block assembly into PRC1.

b) For PHC1, to assess the role of the SAM, mutation of both polymerization interfaces to block both homo and hetero-oligomerization by the SAM should be carried out. To assess interactions with other PRC1 subunits, the HD1 should be deleted in conjunction with SAM polymerization mutant interfaces to block interactions with other PRC1 subunits and endogenous PHCs via the SAM (as in Isono et al., 2013).

c) To test the hypothesis that PRC1 assembly (hetero-oligomerization) is important for condensate formation, knockdown of RNF and PCGF subunits could be used as in the published work. It is more difficult to use domain deletions or mutations for these proteins because the domains involved in protein-protein interactions (Ring finger and RAWUL domains) are important for E3 ligase activity. It

should also be noted that the RAWUL domain of BMI1 is implicated in hetero-oligomerization with the PHC HD1, but also in homo-oligomerization, making interpretation of deletion of this domain difficult.

The previously published work is also not cited precisely cited—for example, the authors state “Purified Cbx2 has previously been shown to undergo concentration-dependent phase separation in vitro (18, 19)”, implying that this observation was not tested in vivo. Yet both of these publications investigate Cbx2-driven condensates in vivo, and identify a critical role for a charged IDR in condensates.

2) The authors have chosen not to mention noncanonical (nc) PRC1. While their experiments indeed focus on cPRC1, I think it is difficult to interpret experiments with RNF2 (which is essential for both cPRC1 and ncPRC1) as being solely due to cPRC1 effects. The images provided suggest that the pattern of corelet-RNF2 is quite distinct from the other subunits. This almost certainly reflects the contribution of ncPRC1 (for example, two ncPRC1 PCGFS, PCGF3 and 5, are implicated in X-chromosome inactivation (e.g. Almeida et al., Science 2017, 10.1126/science.aal2512), and might explain why RNF2 localizes to Xi, while the other tested subunits do not). The authors do not know if ncPRC1 subunits are present in their condensates, or if these complexes can form condensates. This would be testable (for example by using a ncPRC1 PCGF or RYBP in the Corelet system, or using IF or the GFP fusion assay to test colocalization). I am not sure that it is necessary to do these experiments for this paper, since the question of whether ncPRC1 forms/joins condensates could easily be its own study. However, the authors should consider interpretations of RNF2-based experiments more carefully, and alert the reader to this complexity.

3) How was condensate formation assessed? It is my understanding that all of the Corelet-PRC1 subunits are all considered to form condensates upon light activation, yet the patterns all look quite different in the provided images. Can the authors explain how it can be concluded that condensates are formed (quantification and provide quantification and evidence of reproducibility/cell-to-cell variability, as well as an example of a protein that does not form condensates for comparison). It also seems that providing quantitative parameters of the condensates (size, number per cell, etc.) induced by each different subunit and the variability of these parameters is important.

4) In the schematics in Figure 1, certain key domains are lacking (i.e. the HD1 domain of PHC1, and the cbox of CBX2, which mediate assembly into PRC1 (i.e. hetero-oligomerization)), and generic terms like “ZnF” are used for both the FCS domain of PHC1, and the Ring fingers of BMI1 and RNF2, which have completely different functions and quite distinct structures. These schematics also do not capture the analogous structures of BMI1 and RNF2 (i.e. Ring-finger followed by RAWUL). I think the interpretation of the results of co-expression of a GFP-tagged subunit with a corelet-tagged subunit would be easier to navigate if the authors first explained the key interactions known to underly cPRC1 assembly—namely the Ring-Ring interaction between BMI1 and RNF2, and the BMI1-RAWUL—PHC1-HD1, and RNF2-RAWUL-CBX2-cbox interactions. This basic scaffold is consistent with

the interaction results, particularly the low interaction between CBX2 and PHC1, which is predicted from what is known in the literature.

The authors must state precisely which amino acids were included/deleted in each of their constructs. Without this information, it is almost impossible to interpret the results of the structure function analysis.

5) Why was the IDR of BMI1 investigated, but not the well characterized CBX2 IDR, which is implicated in chromatin compaction and condensate formation? This seems critical to making a general conclusion about the role of IDRs. The authors state that “PHC1 is another PRC1 subunit with a native oligomerization domain, but in this case no significant predicted IDR”. Please indicate how this was determined, particularly since the unstructured linker of PHCs that connects the SAM to the HD1 domain has been shown to be functionally relevant for PHC oligomerization (e.g. Robinson et al., 2012, *Biochemistry* doi: /10.1021/bi3004318), and that of PHC homologues. Robinson et al. (2012) (<https://doi.org/10.1021/bi3004318>) conclude that the unstructured linker regulates SAM oligomerization. Since the authors argue later that the SAM is required for the the formation of light-induced Corelet condensates, the unstructured linker as a putative regulator of SAM oligomerization might indeed be significant for interpreting the data. The use of a C-terminal fusion with PHC1 could also affect oligomerization, since C-terminal GFP was previously shown to block clustering of Phc2 (Isono et al., 2013).

6) Some biochemical validation of the constructs used is needed. Simple IP-western blots confirming PRC1 assembly with fusion proteins, and testing if PRC1 assembly is affected by light activation (as done for H2A119Ub) in Fig. S3C) should be provided. This is especially relevant for Fig. 2.

7) Chromatin compaction and H2A119ub experiments should also be done with PHC1 and/or BMI1, which form larger, more obvious condensates. This is especially relevant because Ph SAM-driven phase separation has been shown to enhance H2A119ub in vitro (Seif et al., 2020), and the catalytic activity of RNF proteins (i.e. H2A119Ub) was shown not to be required for chromatin compaction/organization in vivo (Boyle et al., 2020 doi: 10.1101/gad.336487.120). Furthermore, as noted above, condensate formation by PHCs and CBX2 may be distinct, and independent of PRC1, so that the comparison is of considerable interest.

8) The authors state “These repressive histone marks (both H3K27me3 and H2AK119Ub), rather than the condensates themselves, subsequently drive compaction of chromatin.” Yet they have not explicitly tested if compaction can occur in the absence of these modifications. The mutant version of RNF2 does not give clear effects; while the authors’ hypothesis that this is due to endogenous RNF2 may well be correct, this result nevertheless precludes making conclusions about H2A119ub. The double mutant I53A/D56K, which was shown to fully inactivate E3 ligase activity while maintaining PRC1 assembly (Blackledge et al., *Mol Cell*, 2020

<https://doi.org/10.1016/j.molcel.2019.12.001>), or the I53S (Tamburri et al, Mol. Cell 2020 <https://doi.org/10.1016/j.molcel.2019.11.021>) mutant may be more appropriate for these experiments. However, in most studies, substantially reducing H2A119ub requires elimination of endogenous RNF2, so that the ideal experiment would be to deplete the endogenous proteins and then test the role of the mutant (which can form PRC1 but not ubiquitylate H2A). This experiment seems critical to the authors' conclusions.

To test the role of H3K27me3, well-established inhibitors of PRC2 could be used to deplete it.

The finding that chromatin dynamics are distinct from condensate dynamics is interesting; it may be that the authors cannot fully explain the observation at this time. It seems that separating the observation (persistent compaction) from one possible (but not fully tested) interpretation (histone PTMs) could be a better representation of the actual data.

9) The authors claim that “inducing phase separation of PRC1 condensates at any genomic location results in writing of repressive histone marks there” based on the local activation of the Corelet system reported in Figure 4H. While it is apparent from the exemplary images that the degree of colocalization between the RNF2 construct and H2AK119ub increases in the illuminated area, it is not clear from this that the effect applies to “any genomic location”. Based on the image, it seems like the increase in H2AK119ub signal is not uniform throughout the light-activated RNF2 condensate. Thus, there might indeed be differential writing of repressive marks at different genomic locations. More in-depth analysis would be required to back the claim above. It seems likely that regions in constitutive heterochromatin (which contains H3K9me3 instead of H3K27me3) might be less prone to deposition of H2AK119ub by light-induced RNF2 condensates. Given the resolution of the current analysis, the authors should temper their interpretation.

10) Negative controls are consistently absent from the figures; they should be included.

Figure 1: images +/- light of mCherry sspB alone should be included

Figure 2: GFP alone should be used as a negative control, both for images and quantification to determine the “background” colocalization

Figure 3: mCherry-sspB alone (or fused to something that does not form condensates) should be included (as supplemental if it does not fit in the figure).

Figure 4 mCherry-sspB alone should be included for each of the tested correlations (H3K27me3 and H2A119ub). A negative control for the localized illumination should also be included (i.e, confirming that this manipulation does not induce H2A119ub). The authors should also include control experiments confirming that the intense light activation protocol used (both whole cell and focussed) does not inadvertently induce DNA damage. This could be done by staining cells (with

mCherry-sspB alone or fused to a PRC1 subunit) for γ -h2AX. This is particularly important because DNA damage can alter chromatin organization and lead to H2A119ub (e.g. <https://doi.org/10.1016/j.dnarep.2017.06.011>). While it seems unlikely that the 488 wavelength used would induce dsDNA breaks, I think it is important to rule out a contribution from DNA damage.

Figure 5B should include mCherry-sspB only control

11) Figure 3: How was condensates vs. no condensates scored?

Figure 3 F, G—how was the apparent loss of nuclear localization accounted for in the quantification. In 3F, would this pattern (after light activation) be scored as condensates present, or not? There appear to be condensates in the picture.

Additional points:

1) In Figure 1, it might be helpful for the reader if the captions of Fig 1B,F,J,N would clarify that the image IF shows the endogenous protein, and the image OE shows the mCh signal (which is intuitive based on the red color). Since the images in Fig 1C,G,K,O apparently also refer to the mCh signal it might be better to stick to the red color (instead of the white which was used for the immunofluorescence before). Exemplary images of the FRAP experiments could be added to supplement the data presentation. Although the analysis is more or less clear from the figure caption, the graph in Figure 1M is not discussed in the main text. Furthermore, some of the analysis relies on chromatin co-localization but no images of stained DNA are provided. It would make the analysis in Figure 1M more convincing if there would be at least some supplementary data to back the analysis up. This is the case for Figure 1Q in combination with Figure S1A. Nonetheless, it would be helpful to know why the Pearson coefficient that was used in 1M is not used in 1Q, but instead a partition coefficient is introduced that is not explained in more detail.

2) In the paragraph of the methods section describing the phase diagram construction, FCS calibration curves of mCh and GFP were mentioned. It will complement the presented work to include these curves in the supplements because it is central to concentration quantification.

3) The paragraph describing the Pearson correlation coefficient and variance determination indicates that nucleoli and inactivated X-chromosomes were excluded as confounding factors for the degree of co-localization. It is somewhat confusing that the same regions are not excluded in the analysis of the co-expressed PRC1 proteins as if they are not skewing the degree of co-localization in this case. Would it have any effect on the relative “recruitment strengths” shown in Figure 2 if the analysis did not include the nucleoli and inactivated X-chromosomes?

4) Reference 41 needs correction.

Reviewer #3 (Remarks to the Author):

Review of Eeftens et al.

This manuscript is focused on interrogating the relationship between Polycomb condensates, molecular oligomerization, chromatin compaction and epigenetic heritability. The authors use an optogenetic system to nucleate condensates, wild-type and mutant versions of Polycomb condensate components, and advanced imaging and analysis tools. They conclude that hetero-oligomerization of components are required to form condensates, and that such condensates are able to mediate addition of H3K27me3 and H2AK119Ub modifications. Most importantly, they show that although condensates, and by inference phase separation, are able to induce chromatin compaction, but are not required to maintain compaction. There are very interesting findings throughout this paper, and the compaction results are important advances in the field. However, the sole reliance on optogenetic methods, and other issues described below, raises questions about some key interpretations and in particular the direct relevance to endogenous Pc condensates and functions.

General issues:

The optogenetic methods developed by the Brangwynne lab have made major contributions to in vivo analyses of condensate formation and biophysical properties, in part because they are easily manipulated and reversed. However, liquid-like condensate formation depends on networks of multivalent, weak binding interactions among proteins, including associations with 'self' (homo-oligomers) and other proteins/RNAs (hetero-oligomers). Utilizing light-driven protein interactions to drive phase separation creates an artificial system where condensate formation is driven by one valency having affinity strengths and properties that differ from endogenous (not weak?), with unknown effects on formation, properties and regulation in vivo. In short, the relevance to condensates formed from endogenous proteins is unclear, and requires that findings are validated with endogenous proteins and condensates that do not require optogenetic activation. A correlate is that the authors need to show that opto-condensates formed in the absence of the corresponding

endogenous protein(s) display normal properties and Pc functions (ie are the corelets sufficient?). The observation that endogenous Pc proteins can be recruited to the artificial condensates does not address this issue.

In the absence of such validation, the authors need to clearly acknowledge this limitation in descriptions of results and conclusions. In addition, a comparison of the binding affinities for the optogenetic (Corelet) system and endogenous PC proteins should be reported, as well as differences in biophysical properties.

Note that these issues do not affect the conclusions about compaction and maintenance of epigenetic states. These are ok because they are self-contained and independent of how the condensates are formed; the authors clearly show that chromatin compacts upon activation of Pc Corelets, and continues through epigenetic marks after condensate dissolution. It would be better to demonstrate the same is true for endogenous condensates (eg degrade or RNAi endogenous Pc proteins and monitor compaction), but the point is still made effectively.

Specific issues:

levels of expression/overexpression among homo and hetero partners need to be compared

2) p 6. "Thus, the PRC1 Corelet condensates recapitulate key features of endogenous PRC1 complexes, and thus represent light-activatable, amplified versions of endogenous PRC1 condensates."

In part true, but can't really make this conclusion unless endogenous condensates are characterized before light activation....even completely artificial condensates that contain Pc components will of course recruit other components

3) p 6 "However, the phase-diagrams of the two are nearly identical, indicating that the tendency for intracellular phase-separation is not driven by the IDR (Figure 3B,D)." ^[1]_[SEP]Doesn't this reflect dominance of Corelet in driving condensation? Is the IDR essential for endogenous condensates?

Similarly p 7 "while IDRs appear largely dispensable, hetero-oligomerization domains are essential for multicomponent PRC1 condensate formation."

Is that only under these conditions, where corelets alter the balance between valencies? i.e. would the IDR be less dispensable in the wt protein?

4) p7 . "Upon light activation, there was no change in the H3K27me3 pattern,..."

No change means what here? Not the same cell, since fixed, so what specific parameter (intensity, distribution, etc) is referred to not changing? Do not really look the same....smaller K27me3 in OFF. How rule out recruitment altering the K27 pattern? Finally, how much OE?

Also formally this could result from degradation of protein that is not bound to K27me3....do the total levels change upon activation? Is the intensity over K27 the same or increased...hard to tell from the images. PCC hides recruitment specifically to K27 sites by penalizing for signal elsewhere....want to also see quantitation of intensities at K27 sites only

5) p8 "(Figure 4B, S3A,B), consistent with Cbx2WT condensates colocalizing with H3K27me3 marks due to direct Cbx2 reading of these marks."

Are the authors saying that mutant is getting recruited due to interactions with endogenous CBX2? What happens if you delete endogenous CBX2...seems important since in all expts measuring behaviors of constructs in endogenous background, thus not characterizing constructs behavior in isolation. Same applies to the other proteins, eg RNF2 etc...delete endogenous.

6) p8 "However, upon activation and condensation of Cbx2 condensates, H2AK119Ub marks begin appearing over time, at the same location where Cbx2 condensates form (Figure 4C)."

Not convincing....there are no corresponding large blobs of CBX2 in ON, conversely many with equivalent CBX2 levels that do not have as significant or any Ub. Seems like another case of endogenous CBX2 playing an important role?

7) Discussion. Many conclusions need to be qualified with respect to the issues of direct relevance of artificial otto-condensates to endogenous, as well as novelty of some findings.

This statement is novel and well-supported by the last pieces of data presented.

"Using the powerful spatiotemporal control of this system, we demonstrate that PRC1 condensation can induce chromatin compaction, but sustained phase separation is dispensable for maintenance of the compacted state."

However this conclusion is not proven. p12

"These repressive histone marks (both H3K27me3 and H2AK119Ub), rather than the condensates themselves, subsequently drive compaction of chromatin."

It is likely that the marks can maintain compaction only in the presence of the appropriate readers (and writers), even if not condensates per se, since they are required for replenishment during DNA and chromatin replication. Further, compaction could be maintained due to cross linking

components initially recruited by the condensates, that retain association with DNA after condensate dissolves.

REVIEWER COMMENTS

Reviewer #1 (Remarks to the Author):

The present work “Epigenetic marks as a time integral over prior history of Polycomb phase separation” reports that Polycomb Repressive Complex 1 (PRC1) condensates drive chromatin compaction but dispensable for maintaining this compaction. The authors sequentially demonstrated or observed: 1) Core subunits of PRC1 can phase separate in an optogenetic corelet system. 2) PRC1-subunit condensates can recruit other core subunits. 3) Oligomerization domains are responsible for condensate formation. 4) Cbx corelet condensates are localized in H3K27me3 by direct reading, and drive H2AK119Ub writing by recruiting RNF2 proteins into condensates. In addition, RNF2 corelet condensates recruit Cbx subunits, are localized around H3K27me3 and progress H2AK119Ub writing. 5) Cbx corelet condensates drive chromatin compaction but dispensable for maintaining chromatin compaction. These observations provide important findings regarding the relations between Polycomb phase separation, histone modification, and finally chromatin compaction.

We thank the reviewer for emphasizing how our work provides important insights into the fields of phase separation and chromatin organization.

There are, however, several concerns in this manuscript. For example, while it is stated that Polycomb phase separation can induce chromatin compaction but not for maintenance of the compacted state, RNF2 condensates showed only a slight compaction. Therefore, I recommend addressing following issues before considering publication of this work in Nature Communications.

1. In Figure 3, the idea of ‘hetero-oligomerization’ is not clear. What are the components for hetero-oligomerization? (ferritin & oligomerizing domain? Or multiple oligomerizing domains?). If it is hetero-oligomerization of Corelet & PRC1 ODs, then it must be carefully discussed when using observed data for explaining chromatin compaction, since this is not really natural conditions in cells.

We thank the reviewer for pointing out ambiguity in our discussion, which we have addressed by modifying the text. The Corelet system oligomerizes sspB-fused proteins onto Ferritin-iLID cores. This mimics the native oligomerization occurring in PRC1 components and many other phase-separating proteins (e.g. see Sanders et.al. Cell 2020); oligomerization reduces the configurational entropy, which is akin to effectively turning down the temperature. Thus, if the Corelet fused protein is prone to condensate formation, condensates will form, always in a manner dependent on the degree of oligomerization and the concentrations of protein and core. We can map these ratios for each protein with phase diagrams. What the phase diagrams in Figure 3 reveal, is that there is are certain concentrations at which PRC1-subunits form condensates, and concentrations in which no condensates form. If we make changes to the protein, for example cut out the native oligomerization domain, and assess the phase behavior at the same concentration, this is indicative of the condensate-forming ability of that particular protein. For example, when we cut off the native oligomerization domain, such that the only oligomerization is that provided by the Corelet, condensates no longer form at concentrations at which the full-length protein does. Note that sspB-fused proteins are still recruited to the Ferritin-iLID cores, but because the protein lost its ability to bind to other PRC1 components (“hetero-oligomerize”) through its native oligomerization domain, we no longer see condensates. Mapping the phase-diagram for different proteins and domain truncations reveals that the native oligomerization domains in Bmi1 and PHC1 are essential for condensate formation. We’ve adjusted the text to try to make these concepts clearer to the reader.

We use the formation of corelets to probe downstream consequences, such as repressive histone mark writing and chromatin compaction. We show that, for example, formation of Cbx2 corelets results in writing of repressive histone marks and compaction (Figures 3 and 4). However, as shown by the mutated version of Cbx2 (F12A) and other PRC1 proteins, this is not the case for all proteins, indicating that the specific composition of the corelet, and not its mere presence, is causing these specific effects.

2. Relative co-localization is really important to explain many data, particularly for those in Figure 4 and related supporting figures (e.g. Figure 4B, Figure S3A,B). Overlap images will be useful, or can we have more quantitative data on these co-localization data?

We thank the reviewer for this suggestion, and have now added merged images of the fluorescence channels to illustrate co-localization. The quantification of colocalization is illustrated by the Pearson Correlation Coefficient, as shown in Figure 4C and 4E for example. A higher PCC illustrates a higher degree of colocalization.

3. Considering the error bars, I cannot find a clear H2AK119Ub signal increase by light ON in the western blot data (Figure S3C). The PCC data of Figure 4C, however, show clear increases by light activation. Based on these data, it looks like that H2AK119Ub signals are re-distributed by Cbx2-Corelets rather than increased. This issue must be clarified.

The reviewer is correct that although the Western Blot indicates that the total amount of histone modified with H2AK119Ub increases, this increase was not sufficient to establish significance. This is suggestive of Cbx2 introducing new repressive marks, rather than redistributing them. As we are dependent on immunofluorescence for labeling histone marks, we cannot reveal differences in the same cell before and after activation. We have modified our statement on this data to clarify.

4. Since many data contain both cell imaging with fused fluorescent proteins (expressed proteins only) and IF imaging with antibodies (expressed proteins and endogenous proteins), it is very important to understand relative levels of expressed proteins and endogenous proteins. In addition, protein expression levels often largely vary cell-to-cell. Please provide these data and discuss the effects of endogenous proteins in all data explanation (although the author did explain the effects of endogenous proteins for some data).

The expression level of our corelet proteins varies from cell to cell, which is desirable for quantitatively determining the concentrations required for phase separation (i.e. mapping phase diagrams). We have assessed the average overexpression of all PRC1 proteins with western blot, and determined that with the exception of Cbx2, the average overexpression level is <5 fold (see Figure S1B). We have commented on this effect at various places in the manuscript.

For example, to better understand Figure S3E (RNF2_D56K data), where this mutation did not affect Ub patterns much, relative levels of endogenous RNF2 will be important. Ideally, one would like to directly engineer the endogenous RNF2 (and others) gene to fuse various proteins.

The reviewer is correct: there is endogenous RNF2 present in the experiment where we express RNF2_D56K condensates. These condensates likely recruit the endogenous RNF2 and other PRC1 components, such as RNF1. Endogenous RNF1 and 2 proteins also have the ability to write the same H2AK119Ub mark. While we agree that endogenous tagging is generally a desirable strategy for addressing these confounding effects, unfortunately, each protein of the PRC1-multiprotein complex has multiple homologues. Thus, endogenous tagging of a single protein would not overcome the limitation of knowing the contribution of endogenous factors.

5. Figure S4FG, where 'RNF2-Corelets only exhibit a modest triggering of compaction' cannot be explained by the present model. RNF2-Corelets behave (processes like H3K27Me as well as H2AK119Ub) just like Cbx2-Corelets as shown in Figure 4. In addition, RNF2 can recruit other PRC1 components most effectively as shown in Figure 2. Based on these data, RNF2-Corelets must also trigger chromatin compaction as efficient as Cbx2-Corelets. The authors hypothesize this is due to the limited size of RNF2 condensates. However, there is no experimental evidence for this statement. Maybe, condensate sizes could be varied by controlling corelet valency or protein expression levels to examine this statement. In addition, corelet condensates of other two proteins, Bmi1 & PHC1, among PRC1 components must also be examined for their influence on chromatin compaction to support the present model.

We hypothesize that RNF2 only has a modest effect on chromatin compaction because RNF2 corelets are small and coat a smaller portion of the chromatin than Cbx2. There are several reasons for this. First, a recent paper from our lab showed that growth of condensates in the nucleus is significantly slowed by the constraining chromatin network (Lee et al, Nature Physics 2021). Indeed, RNF2 corelets are bound to chromatin, which makes them less likely to coalesce. We can therefore not simply grow condensates larger by tuning the expression level. Furthermore, as we show in Figure 1, a significant amount of RNF2 partitions onto the inactive X-chromosomes. We do not fully understand this strong partitioning, but it is likely due to RNF2's involvement in non-canonical PRC1, and it means there is less RNF2 available to bind elsewhere on the chromatin, resulting in smaller droplets and less compaction. We therefore attribute the modest effect on compaction to the smaller amount of condensate associated chromatin. We have now also measured the effect of Bmi1 and PHC1 condensates and we have added this data to supplementary figure S4. We also added a section with simulation data to that supports our hypothesis that posttranslational modifications can be responsible for maintenance of compaction.

6. The variance determination procedure was only briefly described. Please provide a more detail method for variance determination in Figure 5 and Figure S4.

We have described the procedure in more detail in the methods section.

Reviewer #2 (Remarks to the Author):

In this paper, Eeftens and colleagues use cell imaging and optogenetics to analyze the composition of condensates induced by clustering of PRC1 subunits (through the recently developed Corelet system) and the effect of condensates on H2A119ub and chromatin organization. This implementation of the Corelet system is

the first report of targeted optogenetic manipulation of PRC1 condensate formation.

From the results of these experiments, they draw three main conclusions: 1) condensate formation is driven by hetero-oligomerization; 2) condensates can “read” and “write” histone modifications (H3K27me3 and H2A119ub, respectively); 3) chromatin compaction is not a direct result of phase separation but a secondary result of histone modifications so that it is not required to maintain chromatin compaction. The authors suggest an interesting model in which the persistence of chromatin compaction/histone modification after dissolution of condensates could allow the effects of cycles of condensate formation to be integrated over time to maintain a repressive chromatin state.

The authors have applied an innovative method for inducing condensate formation, which is especially powerful for studying the dynamics of condensates and how they relate to chromatin organization. While the low resolution of the methods used to analyze chromatin effects make it difficult to connect these findings to gene-specific regulation by PRC1, I think the findings and perspective will be of considerable interest to the field. However, I have several major conceptual and technical concerns with respect to the key conclusions. I think that some additional experiments, additional controls, and additional quantification are required, as is additional information about the details of the experiments. The new results should also be interpreted more precisely with respect to published work on PRC1 (particularly work on Cbx2 from the Ren and Kingston groups, and on Phc2 from Koseki).

We thank the reviewer for emphasizing that our methods are innovative and powerful, and that our findings will be of interest to the field.

Major concerns:

1) The authors conclude that hetero-oligomerization drives formation of condensates but this is not clearly supported by their data. The literature suggests an alternative model, namely that both CBX2 and PHC1 can form condensates in cells that do not depend on PRC1 assembly (but can include PRC1 subunits). None of the experiments provided in this manuscript refute this interpretation. Previous work on both Cbx2 (Ren and Kingston labs) and Phc2 (Koseki lab) has demonstrated that both of these proteins can form clusters/condensates in cells without interacting with other PRC1 subunits. For Cbx2, this was shown by analyzing condensates in cells lacking Ring1B/Ring1A (no PRC1 formed) as well as those lacking Bmi1/Mel18 (no canonical PRC1 formed). For Phc1 (Isono et al., Dev. Cell, 2013 doi: 10.1016/j.devcel.2013.08.016.), clustering depends on the polymerization interfaces of the SAM, but not on the HD1 domain that mediates assembly into PRC1.

We thank the reviewer for these comments. We do not think that the data in the literature is in conflict with the data and physical picture we present here. Indeed, our findings show that oligomerization of individual PRC1 subunits can drive condensate assembly. Our findings that the synthetic PRC1 condensates can recruit other PRC1 components through hetero-oligomerization are not inconsistent with the papers cited above, however we realize that the language we used may be mis-interpreted and so we have altered the text to avoid this confusion.

These strategies to test the role of interactions with other PRC1 subunits should be applied in the Corelet system:

a) For CBX2, deletion of the cbx domain should block assembly into PRC1.

We agree with the reviewer that this would be an interesting truncation, as it would reveal interactions of Cbx2 with nucleosomes and other PRC components. It would, for example, be interesting if Cbx2_deltaCbox could still be recruited to corelets formed by other PRC components. We therefore attempted to express Cbx2_deltaCbox-GFP (Cbx2^{aa1-490}) in our cell lines. Unfortunately, we were unable to achieve significant expression of this protein in the nucleus. This can have several reasons. It could be that the deletion of this region results in folding artifacts and/or toxicity, or failure to localize in the nucleus. Interestingly, although there is in vitro literature pointing towards the relevance of this domain for interaction with other PRC components, we could not find any literature expressing a delta-Cbox protein in live human cells. Thus, we speculate that others have also encountered this expression issue.

b) For PHC1, to assess the role of the SAM, mutation of both polymerization interfaces to block both homo and hetero-oligomerization by the SAM should be carried out. To assess interactions with other PRC1 subunits, the HD1 should be deleted in conjunction with SAM polymerization mutant interfaces to block interactions with other PRC1 subunits and endogenous PHCs via the SAM (as in Isono et al., 2013).

The reviewer is correct that these domains are important for oligomerization of PHC1. Indeed, we have deleted the SAM domain (PHC1^{ΔOD}) to illustrate this in our data. In Figure 3JKL, we show that deletion of the SAM domain prevents PHC1 from forming condensates in the Corelet system. This is consistent with the results in Isono et al, where the authors show that the SAM domain is required for clustering. These findings are also confirmed by a recent paper by Seif et al. Furthermore, Isono et al show that a point mutation in the SAM domain abolished recruitment to RNF2 clusters. Also consistent with this, we show that recruitment of PHC1^{ΔOD} fused to

GFP is decreased for RNF2 corelets. We elaborate on this result by showing that PHC1^{ΔOD} is also no longer recruited to Bmi1 and Cbx2 corelets. However, in our system, expressing PHC1^{ΔOD} does not abolish the ability of Bmi1, Cbx2 and RNF2 to form condensates.

The literature the reviewer refers to shows that 1) deletion of the SAM domain of Phc1 blocks oligomerization and the ability of Phc2 to form condensates. We note that our findings are consistent with this result. Furthermore, expressing Phc2 with defective SAM domain captured other PRC1 components, as they could still bind the intact HD1 domain. 2) Deletion of the HD1 domain freed up these other components, thereby restoring their ability to form condensates. As mentioned, after deleting the SAM domain, in our system, we do not lose the ability of condensate formation by other PRC1 components. This indicates there is sufficient Corelets protein and endogenous binding partners, so we conclude that cluster-defective PHC1 is not interfering with the clustering. As we already see no recruitment of PHC1_{deltaSAM} to other corelets, deletion of HD1 would not change this effect. Note that the above-mentioned study was done in mouse fibroblasts with Phc2 and not Phc1.

c) To test the hypothesis that PRC1 assembly (hetero-oligomerization) is important for condensate formation, knockdown of RNF and PCGF subunits could be used as in the published work. It is more difficult to use domain deletions or mutations for these proteins because the domains involved in protein-protein interactions (Ring finger and RAWUL domains) are important for E3 ligase activity. It should also be noted that the RAWUL domain of BMI1 is implicated in hetero-oligomerization with the PHC HD1, but also in homo-oligomerization, making interpretation of deletion of this domain difficult.

We agree with the reviewer that it is difficult to disentangle individual contributions of PRC1 subunits to oligomerization and condensates formation. As the reviewer states, all subunits have multiple homologues. In order to see if, for example, PHC1 is required for condensate formation, knock down of one subunit could result in substitution by another. We therefore feel that knocking down all known PRC1 subunits is beyond the scope of this paper. We have adjusted our phrasing to acknowledge that our results suggest hetero-oligomerization as the driving force behind condensate formation (as is also suggested in the literature), but that our methods are not sufficient to pinpoint the exact contributions of individual proteins.

The previously published work is also not cited precisely cited—for example, the authors state “Purified Cbx2 has previously been shown to undergo concentration-dependent phase separation in vitro (18, 19)”, implying that this observation was not tested in vivo. Yet both of these publications investigate Cbx2-driven condensates in vivo, and identify a critical role for a charged IDR in condensates.

We thank the reviewer for pointing this out - we have changed our phrasing to better describe this published work.

2) The authors have chosen not to mention noncanonical (nc) PRC1. While their experiments indeed focus on cPRC1, I think it is difficult to interpret experiments with RNF2 (which is essential for both cPRC1 and ncPRC1) as being solely due to cPRC1 effects. The images provided suggest that the pattern of corelet-RNF2 is quite distinct from the other subunits. This almost certainly reflects the contribution of ncPRC1 (for example, two ncPRC1 PCGFs, PCGF3 and 5, are implicated in X-chromosome inactivation (e.g. Almeida et al., Science 2017, 10.1126/science.aal2512), and might explain why RNF2 localizes to Xi, while the other tested subunits do not). The authors do not know if ncPRC1 subunits are present in their condensates, or if these complexes can form condensates. This would be testable (for example by using a ncPRC1 PCGF or RYBP in the Corelet system, or using IF or the GFP fusion assay to test colocalization). I am not sure that it is necessary to do these experiments for this paper, since the question of whether ncPRC1 forms/joins condensates could easily be its own study. However, the authors should consider interpretations of RNF2-based experiments more carefully, and alert the reader to this complexity.

We agree with the reviewer that we have not addressed the contribution of ncPRC1 in this work, and we acknowledge the role of RNF2 in ncPRC1 and X-chromosome inactivation. We also agree that studying condensate formation of ncPRC1 components is interesting, but that this would go beyond the scope of this work. We have included the suggested discussion on nc/cPRC1 complexity and the role of RNF2.

3) How was condensate formation assessed? It is my understanding that all of the Corelet-PRC1 subunits are all considered to form condensates upon light activation, yet the patterns all look quite different in the provided images. Can the authors explain how it can be concluded that condensates are formed (quantification and provide quantification and evidence of reproducibility/cell-to-cell variability, as well as an example of a protein that does not form condensates for comparison). It also seems that providing quantitative parameters of the condensates (size, number per cell, etc.) induced by each different subunit and the variability of these parameters is important.

All tested PRC1 subunits form a punctate pattern upon activation in the Corelet system. We emphasize that this is characteristic of the PRC1 proteins, and not the corelet system. For example, mCh-sspB alone does not show a punctate pattern (Supplemental Figure 1). A binary determination of whether or not cells were forming condensates was determined by two independent observers, one of whom was blinded to experimental conditions. The assessments of the two observers were consistent in nearly all cases. The cells on which observers disagreed were excluded from the results. We have clarified this procedure in the methods. Whether or not condensates form is dependent on concentration of the proteins, and indeed there is cell-to-cell variability in condensate forming behavior. Supplemental Figure 1 includes additional images (for the purpose of demonstrating colocalization) of each protein to give the reader an idea of this variability. For an example of proteins that do not form condensates under the tested conditions, we refer the reader to Figure 3, in which we demonstrate protein truncations that abolish condensate forming capabilities. We have also added mCh-sspB alone for comparison in Supplemental Figure 1.

In order for us to comment on the size and number of condensates per cell, we would have to segment out the condensates. Due to their small size which in many cases is comparable to the diffraction limit, this has been technically difficult, and it introduces a lot of user bias into the system. For example, Cbx2 condensates have a strong preference to colocalize with the chromatin. Adjacent condensates are often small and close together, making it difficult to discern if they are separate. Due to these complications, we feel that quantifying size will not be particularly helpful in this case.

4) In the schematics in Figure 1, certain key domains are lacking (i.e. the HD1 domain of PHC1, and the cbox of CBX2, which mediate assembly into PRC1 (i.e. hetero-oligomerization)), and generic terms like “ZnF” are used for both the FCS domain of PHC1, and the Ring fingers of BMI1 and RNF2, which have completely different functions and quite distinct structures. These schematics also do not capture the analogous structures of BMI1 and RNF2 (i.e. Ring-finger followed by RAWUL). I think the interpretation of the results of co-expression of a GFP-tagged subunit with a corelet-tagged subunit would be easier to navigate if the authors first explained the key interactions known to underly cPRC1 assembly—namely the Ring-Ring interaction between BMI1 and RNF2, and the BMI1-RAWUL—PHC1-HD1, and RNF2-RAWUL-CBX2-cbox interactions. This basic scaffold is consistent with the interaction results, particularly the low interaction between CBX2 and PHC1, which is predicted from what is known in the literature.

We thank the author for this suggestion, and we agree that our results are consistent with literature. We have adjusted the domains in Figure 1 to more precisely represent the literature nomenclature. We also elaborated on the interactions in the revised manuscript as the reviewer suggests.

The authors must state precisely which amino acids were included/deleted in each of their constructs. Without this information, it is almost impossible to interpret the results of the structure function analysis.

We have included this information in the plasmid list in the Supplementary Information section.

5) Why was the IDR of BMI1 investigated, but not the well characterized CBX2 IDR, which is implicated in chromatin compaction and condensate formation? This seems critical to making a general conclusion about the role of IDRs.

The reviewer is correct, the IDR of Cbx2 has been shown to play a role in phase-separation of Cbx2. In particular, the charged residues within in this IDR of the mouse Cbx2 (Plys et al, Genes and Development 2019). For technical reasons, we were unable to make a construct with the point mutations that are required to do the equivalent experiment for the human protein in the corelet system. We agree with the reviewer that we can therefore not make a general statement about the importance of IDRs in PRC1 proteins. We have adjusted our phrasing to make it clear we can only speak on the IDR of Bmi1.

The authors state that “PHC1 is another PRC1 subunit with a native oligomerization domain, but in this case no significant predicted IDR”. Please indicate how this was determined, particularly since the unstructured linker of PHCs that connects the SAM to the HD1 domain has been shown to be functionally relevant for PHC oligomerization (e.g. Robinson et al., 2012, Biochemistry doi: /10.1021/bi3004318), and that of PHC homologues. Robinson et al. (2012) (<https://doi.org/10.1021/bi3004318>) conclude that the unstructured linker regulates SAM oligomerization. Since the authors argue later that the SAM is required for the the formation of light-induced Corelet condensates, the unstructured linker as a putative regulator of SAM oligomerization might indeed be significant for interpreting the data. The use of a C-terminal fusion with PHC1 could also affect oligomerization, since C-terminal GFP was previously shown to block clustering of Phc2 (Isono et al., 2013).

We have used an open source disorder predictor to look for disordered domains (<https://iupred2a.elte.hu/>). In the case of PHC1, there are no long stretches of high disorder score. Generally, short linkers, while they can be important for function and typically conformational heterogenous, are usually not referred to as “disordered

domains". We acknowledge that this distinction is not very quantitative, so we have removed the statement on PHC1 lacking an IDR from our manuscript. There may be shorter stretches that are relevant and contributing to condensate formation, but with the exception of the linker the reviewer refers to, we are unaware of previous reports in literature.

We thank the reviewer for pointing out that the linker region in between HD1 and SAM domains might regulate oligomerization. We did not test this contribution specifically in our assays. As mentioned by the reviewer, we truncate the SAM domain but leave the HD1 and linker domain. This is sufficient to abolish condensate formation and recruitment of PHC1 to other PRC1 condensates. C-terminal fusion of full-length PHC1 with GFP or mCh-sspB does not abolish oligomerization or recruitment of PHC1 into PRC1-condensates (Figures 1 and 2 and supplement). We have adjusted our phrasing to make it clear we cannot speak for all IDRs in PRC1 proteins.

6) Some biochemical validation of the constructs used is needed. Simple IP-western blots confirming PRC1 assembly with fusion proteins, and testing if PRC1 assembly is affected by light activation (as done for H2A119Ub) in Fig. S3C) should be provided. This is especially relevant for Fig. 2.

We agree that it is important to address the ability of our light-activatable PRC1 condensates to interact with other PRC1 components. This is a key reason why we extensively assess recruitment into condensates in Figure 2, and we also assess the changes in recruitment when we make truncations that we know from literature affect interactions. As the reviewer states above, these results are consistent with literature and we recapitulate the interactions known. This provides strong validation that our light-activatable condensates behave as expected.

7) Chromatin compaction and H2A119ub experiments should also be done with PHC1 and/or BMI1, which form larger, more obvious condensates. This is especially relevant because Ph SAM-driven phase separation has been shown to enhance H2A119ub in vitro (Seif et al., 2020), and the catalytic activity of RNF proteins (i.e. H2A119Ub) was shown not to be required for chromatin compaction/organization in vivo (Boyle et al., 2020 doi: 10.1101/gad.336487.120). Furthermore, as noted above, condensate formation by PHCs and CBX2 may be distinct, and independent of PRC1, so that the comparison is of considerable interest.

We have added colocalization of Bmi1 and PHC1 with H3K27me3 and H2AK119Ub to Figure S3. We do not see significant overlap. Note that this is a different metric than used in Seif et al, in which they show an increase in H2AK119ub upon Ph overexpression in drosophila cells. Our results indicate that chromatin compaction is driven by histone modification through chromatin interaction of RNF2 and Cbx2 condensates. However, we do not disagree with the reviewer (and Boyle et al) that changes in the chromatin architecture can be regulated separately from chromatin compaction and catalytic activity. We elaborate on this in the discussion of the revised manuscript.

8) The authors state "These repressive histone marks (both H3K27me3 and H2AK119Ub), rather than the condensates themselves, subsequently drive compaction of chromatin." Yet they have not explicitly tested if compaction can occur in the absence of these modifications. The mutant version of RNF2 does not give clear effects; while the authors' hypothesis that this is due to endogenous RNF2 may well be correct, this result nevertheless precludes making conclusions about H2A119ub. The double mutant I53A/D56K, which was shown to fully inactivate E3 ligase activity while maintaining PRC1 assembly (Blackledge et al., Mol Cell, 2020 <https://doi.org/10.1016/j.molcel.2019.12.001>), or the I53S (Tamburri et al, Mol. Cell 2020 <https://doi.org/10.1016/j.molcel.2019.11.021>) mutant may be more appropriate for these experiments. However, in most studies, substantially reducing H2A119ub requires elimination of endogenous RNF2, so that the ideal experiment would be to deplete the endogenous proteins and then test the role of the mutant (which can form PRC1 but not ubiquitylate H2A). This experiment seems critical to the authors' conclusions. To test the role of H3K27me3, well-established inhibitors of PRC2 could be used to deplete it. The finding that chromatin dynamics are distinct from condensate dynamics is interesting; it may be that the authors cannot fully explain the observation at this time. It seems that separating the observation (persistent compaction) from one possible (but not fully tested) interpretation (histone PTMs) could be a better representation of the actual data.

Our data indeed shows that sustained compaction does not require the presence of condensates, but that the presence of condensates leads to the writing of repressive histone marks. This supports our hypothesis that it is the posttranslational modifications that are leading to compaction, a concept that is further supported in the revised manuscript through the addition of new simulation data. However, we agree with the reviewer that testing if compaction can occur in the absence of repressive modifications would provide an additional means of examining the hypothesis that it is the repressive histone marks that are driving compaction. Unfortunately, these experiments are technically challenging, as this would require removal of both H2AK119Ub and H3K27me3, which may be at least partially achieved by double knockout of both RNF proteins and PRC2 components. We thank the reviewer for understanding that we may not be able to fully explain the observation at this time. In the revised manuscript, we have adapted the phrasing to suggest this as a future experiment, and toned down our claims to clarify we cannot conclude this definitively. Furthermore, we added simulation data to strengthen our

point. The simulation data shows the same pattern as our experimental results and therefore suggests that indeed, posttranslational modifications can be sufficient for sustained compaction.

9) The authors claim that “inducing phase separation of PRC1 condensates at any genomic location results in writing of repressive histone marks there” based on the local activation of the Corelet system reported in Figure 4H. While it is apparent from the exemplary images that the degree of co-localization between the RNF2 construct and H2AK119ub increases in the illuminated area, it is not clear from this that the effect applies to “any genomic location”. Based on the image, it seems like the increase in H2AK119ub signal is not uniform throughout the light-activated RNF2 condensate. Thus, there might indeed be differential writing of repressive marks at different genomic locations. More in-depth analysis would be required to back the claim above. It seems likely that regions in constitutive heterochromatin (which contains H3K9me3 instead of H3K27me3) might be less prone to deposition of H2AK119ub by light-induced RNF2 condensates. Given the resolution of the current analysis, the authors should temper their interpretation.

The reviewer observes correctly that while there is a clear increase in H2AK119Ub signal upon local activation in a particular area, we do not identify which specific area that is (euchromatin/constitutive or facultative heterochromatin). We agree with the reviewer that we can therefore not fully justify the claim that marks can be written at “any genomic location”. We have adjusted the phrasing to temper this statement.

10) Negative controls are consistently absent from the figures; they should be included.
Figure 1: images +/- light of mCherry sspB alone should be included

We included the requested control in Figure S1.

Figure 2: GFP alone should be used as a negative control, both for images and quantification to determine the “background” colocalization

We added the requested control to Figure 2.

Figure 3: mCherry-sspB alone (or fused to something that does not form condensates) should be included (as supplemental if it does not fit in the figure).

mCherry-sspB by itself is not able to form condensates. We show a qualitative example in Figure S1, and this control has been shown in other papers from our lab as well (Bracha et al 2018, Sanders et al 2020).

Figure 4 mCherry-sspB alone should be included for each of the tested correlations (H3K27me3 and H2A119ub). A negative control for the localized illumination should also be included (i.e, confirming that this manipulation does not induce H2A119ub).

We added the requested control for global activation to the Figure 4 and corresponding supplemental figures. We also added a local activation control to Supplementary Figure 3.

The authors should also include control experiments confirming that the intense light activation protocol used (both whole cell and focussed) does not inadvertently induce DNA damage. This could be done by staining cells (with mCherry-sspB alone or fused to a PRC1 subunit) for γ -h2AX. This is particularly important because DNA damage can alter chromatin organization and lead to H2A119ub (e.g. <https://doi.org/10.1016/j.dnarep.2017.06.011>). While it seems unlikely that the 488 wavelength used would induce dsDNA breaks, I think it is important to rule out a contribution from DNA damage.

We thank the reviewer for this suggestion, and we agree that the connection between DNA damage repair and H2AK119Ub is interesting. We stained for DNA damage with γ H2AX for activated and nonactivated cells. We did not find an increased amount of DNA damage as a result of activation, and therefore conclude that the increased signal in H2AK119Ub is not due to DNA damage induced by the light activation protocol. We added this control to Supplementary Figure 3.

Figure 5B should include mCherry-sspB only control

We have added a mCh-sspB only control, as well as data on Bmi1 and PHC1 to Supplementary Figure 4.

11) Figure 3: How was condensates vs. no condensates scored?

Figure 3 F, G—how was the apparent loss of nuclear localization accounted for in the quantification. In 3F, would this pattern (after light activation) be scored as condensates present, or not? There appear to be condensates in the picture.

See also our response to the reviewer above. We have added more detail on how we score the presence of condensates in the methods section. In the case of loss of nuclear localization, we measure the concentration of the protein in the nucleus for the phase diagram. In the case of Bmi1dOD, we do not see de novo condensate formation at the measured nuclear concentrations (Figure 3FG). As the reviewer correctly remarks, we still see slight growth of pre-existing puncta upon activation. We have mentioned observation in the manuscript text.

Additional points:

1) In Figure 1, it might be helpful for the reader if the captions of Fig 1B,F,J,N would clarify that the image IF shows the endogenous protein, and the image OE shows the mCh signal (which is intuitive based on the red color). Since the images in Fig 1C,G,K,O apparently also refer to the mCh signal it might be better to stick to the red color (instead of the white which was used for the immunofluorescence before).

We have clarified Figure 1 to emphasize which figures are overexpression and which are immunofluorescence. However, we feel that the black and white image shows better contrast, so we have decided to keep this color map for figures C,G,K and O. For consistency and to avoid confusion, we have converted the overexpression images to grayscale as well.

Exemplary images of the FRAP experiments could be added to supplement the data presentation.

We have added examples of FRAP experiments to Supplementary Figure 1.

Although the analysis is more or less clear from the figure caption, the graph in Figure 1M is not discussed in the main text. Furthermore, some of the analysis relies on chromatin co-localization but no images of stained DNA are provided. It would make the analysis in Figure 1M more convincing if there would be at least some supplementary data to back the analysis up. This is the case for Figure 1Q in combination with Figure S1A. Nonetheless, it would be helpful to know why the Pearson coefficient that was used in 1M is not used in 1Q, but instead a partition coefficient is introduced that is not explained in more detail.

We thank the reviewer for bringing this oversight to our attention. We have now referenced this result and analysis to the revised manuscript. We used Hoechst to stain chromatin and assess colocalization with Cbx2. Example images of similar experiments can be found in for example Figure 4. Furthermore, we have alerted the reader to the location of the condensate enrichment, in particular the known heterochromatic regions surrounding nucleoli.

For Figure 1Q, we wanted to demonstrate increased partitioning on the inactivated X-chromosome in particular, in contrast to Figure 1M, in which we are looking the entire nucleus for chromatin localization. As the inactivated X-chromosome is easy to identify, we were comfortable using the partition coefficient. We have added a more detailed description of partition coefficient determination in the revised manuscript.

2) In the paragraph of the methods section describing the phase diagram construction, FCS calibration curves of mCh and GFP were mentioned. It will complement the presented work to include these curves in the supplements because it is central to concentration quantification.

We have included the reference for a previous paper from our lab that describes the methods for obtaining these FCS curves in detail.

3) The paragraph describing the Pearson correlation coefficient and variance determination indicates that nucleoli and inactivated X-chromosomes were excluded as confounding factors for the degree of co-localization. It is somewhat confusing that the same regions are not excluded in the analysis of the co-expressed PRC1 proteins as if they are not skewing the degree of co-localization in this case. Would it have any effect on the relative "recruitment strengths" shown in Figure 2 if the analysis did not include the nucleoli and inactivated X-chromosomes?

It is correct that we exclude nucleoli and inactivated X-chromosome from the colocalization between PRC1 components and H3K27me3 and H2AK119Ub as shown in Figure 2, but we do not for the colocalization in Figure 2. For the experiments in Figure 2, we do not have a reliable marker for either the nucleoli or the X-chromosome. We apologize for this confusion and will clarify this in the methods.

We tried to quantify the effect on recruitment strength as the reviewer suggests. In the case of Cbx2-mCh and RNF2-GFP, both proteins localize to the inactivated X-chromosome, and we can roughly identify nucleoli by looking at mutual exclusion. For this particular example, we quantified the co-localization with both bodies included to be on average 3% higher than for the case in which we exclude them. As this is not a huge difference, and because the X-chromosome is a relevant region for colocalization, we feel confident in leaving it included in the analysis as is.

4) Reference 41 needs correction.

We thank the reviewer for bringing this to our attention. We have made the correction.

Reviewer #3 (Remarks to the Author):

Review of Eeftens et al.

This manuscript is focused on interrogating the relationship between Polycomb condensates, molecular oligomerization, chromatin compaction and epigenetic heritability. The authors use an optogenetic system to nucleate condensates, wild-type and mutant versions of Polycomb condensate components, and advanced imaging and analysis tools. They conclude that hetero-oligomerization of components are required to form condensates, and that such condensates are able to mediate addition of H3K27me3 and H2AK119Ub modifications. Most importantly, they show that although condensates, and by inference phase separation, are able to induce chromatin compaction, but are not required to maintain compaction. There are very interesting findings throughout this paper, and the compaction results are important advances in the field. However, the sole reliance on optogenetic methods, and other issues described below, raises questions about some key interpretations and in particular the direct relevance to endogenous Pc condensates and functions.

We thank the reviewer for emphasizing our findings are interesting and important advances in the field.

General issues:

The optogenetic methods developed by the Brangwynne lab have made major contributions to in vivo analyses of condensate formation and biophysical properties, in part because they are easily manipulated and reversed. However, liquid-like condensate formation depends on networks of multivalent, weak binding interactions among proteins, including associations with 'self' (homo-oligomers) and other proteins/RNAs (hetero-oligomers). Utilizing light-driven protein interactions to drive phase separation creates an artificial system where condensate formation is driven by one valency having affinity strengths and properties that differ from endogenous (not weak?), with unknown effects on formation, properties and regulation in vivo. In short, the relevance to condensates formed from endogenous proteins is unclear, and requires that findings are validated with endogenous proteins and condensates that do not require optogenetic activation. A correlate is that the authors need to show that opto-condensates formed in the absence of the corresponding endogenous protein(s) display normal properties and Pc functions (ie are the corelets sufficient?). The observation that endogenous Pc proteins can be recruited to the artificial condensates does not address this issue. In the absence of such validation, the authors need to clearly acknowledge this limitation in descriptions of results and conclusions. In addition, a comparison of the binding affinities for the optogenetic (Corelet) system and endogenous PC proteins should be reported, as well as differences in biophysical properties.

Note that these issues do not affect the conclusions about compaction and maintenance of epigenetic states. These are ok because they are self-contained and independent of how the condensates are formed; the authors clearly show that chromatin compacts upon activation of Pc Corelets, and continues through epigenetic marks after condensate dissolution. It would be better to demonstrate the same is true for endogenous condensates (eg degrade or RNAi endogenous Pc proteins and monitor compaction), but the point is still made effectively.

We fully agree with the reviewer that condensate formation depends on multivalent binding of proteins and nucleic acids, and that biological condensates are multicomponent (this has been the central point of several recent related papers from our lab, see e.g. Riback 2020 and Sanders 2020). The Corelet system is a way to locally control protein oligomerization, and depending on their self-self and self-other interactions, a condensate will form. This is fully dependent on the protein's properties, and the protein's affinity for self-self and self-other interactions, as there is no light-induced forced interaction between a protein and their binding partners. For example, mCh-sspB alone will not form condensates. Using the Corelet system therefore enables examining the native properties and interactions of proteins in living cells, in their natural environment where they are free to recruit endogenous interaction partners. Condensate formation is therefore dependent on the endogenous interaction network; for example, see Sanders et.al., where the Corelet system is used to build synthetic stress granules, which recapitulate all features of native stress granules, including dependence on heterotypic interactions with RNA.

Polycomb bodies/condensates have been reported in the literature for several decades now. It is widely accepted that PRC proteins form punctate patterns, and that the condensates they form are multicomponent. There are several domains that have been identified to be important for condensate binding behavior, and for reading and writing histone marks. The reviewer requests that we show that these opto-condensates display normal Polycomb properties and functions. We have also gotten this feedback after our first BioRxiv submission. Since

then, we have added immunofluorescence data to show the previously reported punctate patterns in this cell type (Figure 1) and shown our overexpressed proteins form amplified versions of the endogenous pattern. We show that PRC proteins recruit their native binding partners (Figure 2), that interactions with PRC components and nucleosomes depend on the domains reported in literature, and that they read and write the histone marks they endogenously interact with. We have shown that our light-activatable condensates recapitulate essentially every aspect of endogenous condensates as reported in literature. We have highlighted this at several places in the manuscript. Finally, we agree with the reviewer that conclusions on maintenance of epigenetic states are independent of the way condensates are formed.

Specific issues:

levels of expression/overexpression among homo and hetero partners need to be compared

We have assessed the average overexpression of all PRC proteins with western blot, and determined that with the exception of Cbx2, the average overexpression level is <5 fold (see Figure S1B). We have commented on this effect at various places in the manuscript.

2) p 6. "Thus, the PRC1 Corelet condensates recapitulate key features of endogenous PRC1 complexes, and thus represent light-activatable, amplified versions of endogenous PRC1 condensates."

In part true, but can't really make this conclusion unless endogenous condensates are characterized before light activation....even completely artificial condensates that contain Pc components will of course recruit other components

Polycomb condensates have been widely reported in the literature. Endogenous labeling of PRC1 with immunofluorescence shows that condensates are formed in absence of the corelet system. In order to study downstream consequences of condensate formation, we used the corelet system to induce de novo PRC1 condensates. We stand by our statement that these light-induced condensates recapitulate key features of PRC1 complexes, including recruitment of endogenous binding partners. As described in the responses above, there is overwhelming experimental support for this statement.

3) p 6 "However, the phase-diagrams of the two are nearly identical, indicating that the tendency for intracellular phase-separation is not driven by the IDR (Figure 3B,D)." Doesn't this reflect dominance of Corelet in driving condensation? Is the IDR essential for endogenous condensates? Similarly p 7 "while IDRs appear largely dispensable, hetero-oligomerization domains are essential for multicomponent PRC1 condensate formation." Is that only under these conditions, where corelets alter the balance between valencies? i.e. would the IDR be less dispensable in the wt protein?

The Corelet system itself does not drive condensate formation; in driving oligomerization, Corelets represent a kind of "entropic knob" which together with the interaction network of the particular oligomerized protein, may or may not drive phase separation. For example, Corelets formed with mCh-sspB alone will not form condensates (Figure S1A). Truncation of the IDR of Bmi1 does not affect condensate formation, but truncation of the oligomerization domain of Bmi1 does (Figure 3FG). This indicates that it is not the mere presence of the Corelet that is driving the condensate formation, but instead is driven by oligomerization of Bmi1. We agree with the reviewer that while we show the IDR of Bmi1 is not essential for condensate formation, we cannot make a general statement about the importance of IDRs in PRC1 proteins. We have adjusted our phrasing to make it clear we can only speak on the IDR of Bmi1 (see also response above).

4) p7 . "Upon light activation, there was no change in the H3K27me3 pattern,..."

No change means what here? Not the same cell, since fixed, so what specific parameter (intensity, distribution, etc) is referred to not changing? Do not really look the same....smaller K27me in OFF. How rule out recruitment altering the K27 pattern?

The reviewer is correct, due to the fact that we use immunofluorescence on fixed cells as our marker, we cannot compare cells before and after light activation. The overall intensity of H3K27me3 stain appears to be constant before and after activation, but as these are different cells and different samples, it is difficult to compare. We have included a Western Blot in Supplementary Figure 4 to show that the H3K27me3 levels do not appear to change with Cbx2-corelet activation.

Finally, how much OE?

See our response above.

Also formally this could result from degradation of protein that is not bound to K27me3....do the total levels change upon activation? Is the intensity over K27 the same or increased...hard to tell from the images. PCC

hides recruitment specifically to K27 sites by penalizing for signal elsewhere....want to also see quantitation of intensities at K27 sites only

We have included a Western Blot in Supplementary Figure 3 to show that the H3K27me3 levels do not appear to change with Cbx2-corelet activation.

5) p8 "(Figure 4B, S3A,B), consistent with Cbx2WT condensates colocalizing with H3K27me3 marks due to direct Cbx2 reading of these marks."

Are the authors saying that mutant is getting recruited due to interactions with endogenous CBX2? What happens if you delete endogenous CBX2...seems important since in all expts measuring behaviors of constructs in endogenous background, thus not characterizing constructs behavior in isolation. Same applies to the other proteins, eg RNF2 etc...delete endogenous.

All PRC1 proteins have multiple homologues that have similar functions. Deleting of single components would not abolish recruitment to PRC sites, as multiple homologues are able to take over function. Unfortunately, knocking down all ~20 canonical PRC1 proteins to examine their individual contributions goes beyond the scope of this paper.

6) p8 "However, upon activation and condensation of Cbx2 condensates, H2AK119Ub marks begin appearing over time, at the same location where Cbx2 condensates form (Figure 4C)."

Not convincing...there are no corresponding large blobs of CBX2 in ON, conversely many with equivalent CBX2 levels that do not have as significant or any Ub. Seems like another case of endogenous CBX2 playing an important role?

The reviewer is correct that Cbx2 corelets do not form "large blobs", but rather form to their endogenous localization at H3K27me3 sites. We therefore do not expect to find large puncta of de novo H2AK119Ub marks. Rather, we see that the mark follows the Cbx2 corelet localization. We also want to emphasize that the H2AK119Ub areas of larger intensity are inactivated X-chromosomes, which also appear in the unactivated cells.

7) Discussion. Many conclusions need to be qualified with respect to the issues of direct relevance of artificial otto-condensates to endogenous, as well as novelty of some findings.

This statement is novel and well-supported by the last pieces of data presented.

"Using the powerful spatiotemporal control of this system, we demonstrate that PRC1 condensation can induce chromatin compaction, but sustained phase separation is dispensable for maintenance of the compacted state."

However this conclusion is not proven. p12

"These repressive histone marks (both H3K27me3 and H2AK119Ub), rather than the condensates themselves, subsequently drive compaction of chromatin."

It is likely that the marks can maintain compaction only in the presence of the appropriate readers (and writers), even if not condensates per se, since they are required for replenishment during DNA and chromatin replication. Further, compaction could be maintained due to cross linking components initially recruited by the condensates, that retain association with DNA after condensate dissolves.

We indeed hypothesize that the compaction is due to the contribution of histone marks, but the reviewer is correct that while our manuscript is consistent with and strongly supports this hypothesis, other explanations are formally possible. We have adapted the phrasing to emphasize this caveat. In addition, we have added simulations to the manuscript that support our hypothesis.

REVIEWER COMMENTS

Reviewer #1 (Remarks to the Author):

The authors properly addressed all raised issues of the original manuscript. Now, the revised work sufficiently support the model that the authors suggest. Therefore, I recommend publication of this work.

Reviewer #2 (Remarks to the Author):

The authors have addressed each point raised in the initial review. I do not find some of their arguments convincing (particularly the insistence that these effects are dependent on PRC1 formation and hetero-oligomerization and the insistence that H2A119Ub is involved despite showing only correlations, not doing the experiments that would test this, and brushing aside results with RNF2 that seem at odds with this interpretation). Thus, I disagree with the data interpretation (or feel that it goes beyond what is shown), but nevertheless find the observations novel and interesting, and remain willing to be convinced by future experiments. There are still several points that should be addressed/corrected/clarified.

1. The Western blot analysis in Figure S1B indicates that CBX2 is overexpressed by 20x, ≥ 4 x more than any of the other proteins. This should be taken into consideration when interpreting the condensates, particularly since CBX2 is the only protein that seems to compact chromatin. It is possible that given the low resolution of the analysis of chromatin compaction, higher expression levels of the other proteins could also allow chromatin compaction. The very high level of overexpression of CBX2 is also not consistent with its effects reflecting the activity of PRC1. Given the importance of the CBX2F12A mutant for the conclusions, overexpression of this protein should also be analyzed—i.e. to confirm that its effects are due to the mutation and not lower expression.

As a more minor point, the western blots should be shown in standard format (not overlaid on the bar graph) with size markers, and the endogenous and overexpressed bands indicated.

2. Figure 1N The E3 ligase activity is actually mediated by the Ring domain. RNF2 consists of a RING and RAWUL, very similar to BMI1, except without the small C-terminal IDR. This should be corrected. For clarification, Figure 2 of this review nicely shows the configuration of BMI-RING proteins vis-a-vis the E3 ligase activity.

<https://doi.org/10.1016/j.exphem.2016.12.006>

3. Please clarify H2B-miRFP is H2B-miRFP670 in the text and figure legends (otherwise it appears that two red FPs are being used in the same experiment).

4. The construction of the phase diagrams in Figure 3 needs further explanation. If I understand the figure legend, each symbol is a single cell, so that many of the points seem to be represented by a very small number of cells. The number of cells analyzed at each point, and how many did versus did not form condensates should be provided as source data. This is what is written in the methods section:

A standard imaging protocol was used on all cells to avoid variability. All activation protocols were 3 minutes with 2 second intervals. Only cells that were fully in the field of view were considered. The average GFP and mCh fluorescence intensity was determined using the first frame (before activation). Determination of whether or not cells were forming condensates was determined qualitatively by two independent observers, one of whom was blinded to experimental conditions. The assessments of the two observers were consistent in nearly all cases. The few cells on which observers disagreed were excluded from the results. FCS calibration curves were used to determine the mCh and GFP concentrations from the fluorescence intensities as described in Bracha et al (2018). Valence was measured as the ratio of sspB-fused protein to core.

Where does the GFP come from? Can images of GFP be provided to support the ratios? It is my understanding that in this experiment the ILid construct contains GFP (as in Bracha et al., 2018, Fig. 1), but in the other experiments (at least the GFP colocalization ones), it does not? Please state clearly which constructs were used in each experiment (or if possible include a schematic as in Bracha et al. Fig. 1). This is especially confusing in Fig. 3 because the colocalization with GFP-PcG proteins is intermixed with the phase diagrams.

Please move the legends off the graphs as it is not possible for the reader to know if they obscure data points.

5. In figure 5, there is no "H" in the figure. It appears that the bottom part of F should be "G". More importantly, how does the model behave if there is no H2A119Ub?

6. The description of image analysis (which is central to the conclusions of this paper) is not adequate to allow another group to reproduce it. The methods currently state:

Image analysis

All images were analysed using a combination of manual segmentation (ImageJ) and automated segmentation in Matlab.

7. For the description of the variance analysis :

For variance determination, the nucleus was segmented in every frame, with nucleoli excluded.

For each timepoint, the variance in pixel intensities was determined in each channel as:

Where n is the sample size (number of pixels), and i is the individual pixel index, with A_i the value in that pixel, and μ the mean. The variance over time was averaged over the amount of cells specified in the figure legend, and normalized.

What is meant by « the amount of cells specified in the figure legend »--i did not find information about the amount of cells in the figure legends. And how were the data normalized?

8. A note on nomenclature. Typically the human proteins are indicated in all caps (i.e. BMI1, CBX2); as written, the figures/text imply that the mouse proteins (Bmi1, Cbx2) were used. Unless this is the case, it would be good to stay with the standard nomenclature (as was done for RNF2 and PHC1).

9. From the previous review :

1) The authors conclude that hetero-oligomerization drives formation of condensates but this is not clearly

supported by their data. The literature suggests an alternative model, namely that both CBX2 and PHC1 can

form condensates in cells that do not depend on PRC1 assembly (but can include PRC1 subunits). None of the

experiments provided in this manuscript refute this interpretation.

Previous work on both Cbx2 (Ren and Kingston labs) and Phc2 (Koseki lab) has demonstrated that both of these

proteins can form clusters/condensates in cells without interacting with other PRC1 subunits. For Cbx2, this was

shown by analyzing condensates in cells lacking Ring1B/Ring1A (no PRC1 formed) as well as those lacking

Bmi1/Mel18 (no canonical PRC1 formed). For Phc1 (Isono et al., Dev. Cell, 2013 doi:

10.1016/j.devcel.2013.08.016.), clustering depends on the polymerization interfaces of the SAM, but not on the

HD1 domain that mediates assembly into PRC1.

We thank the reviewer for these comments. We do not think that the data in the literature is in conflict with the

data and physical picture we present here. Indeed, our findings show that oligomerization of individual PRC1

subunits can drive condensate assembly. Our findings that the synthetic PRC1 condensates can recruit other

PRC1 components through hetero-oligomerization are not inconsistent with the papers cited above, however we

realize that the language we used may be mis-interpreted and so we have altered the text to avoid this confusion.

I do not see that this response addresses the criticism. Particularly for CBX2, i do not find evidence supporting a requirement for PRC1 in condensate formation. It seems equally possible that most of what is observed is CBX2 alone, driven by the high level of overexpression (see note regarding sFig. X) It is unfortunate that the authors refuse to do simple IP experiments (which could be doen with the reagents in hand) to at least begin to address what fraction of their CBX2 is in PRC1 versus on its own.

Minor points:

1. Fig. S2B PHC1 Δ OD-GFP is labelled as " Δ OD" in legend but " Δ SAM" on image
2. In the source data for Figure S3G, the blot is labelled "H2AK119Ub", but the title is "Quantification Western Blot H3K27me3"

Reviewer #3 (Remarks to the Author):

This reviewer is satisfied with the majority of the responses and revisions, and fully supports publication of this manuscript.

However, it is worth clarify points concerning interpretations (not the data or primary conclusions, which are sound). My hope is that the authors will step back and consider that there are in fact limitations to the system and findings, as well as alternative explanations for the results, that are worth mentioning in the Discussion. Consider this an academic discussion, not a critique of the content or validity of the paper, nor a requirement for publication.

It is exciting and appropriate to claim that the condensate per se is not required to maintain compaction in this system. However, the authors should acknowledge that it could just as easily require both the marks and the reader proteins, even if they are not forming a condensate (eg compaction via local crosslinks mediated by the readers). The emphasis on the sufficiency of histone

marks for chromatin compaction is certainly strengthened by the modeling, but it is not validated experimentally. That requires the demonstration that compaction occurs in the presence of the marks after eliminating or interfering with the presence or function of the reader proteins.

In addition, the authors' vigorous defense of the Corelet system is understandable and appropriate; this and other optosystems developed in the Brangwynne lab and elsewhere are powerful. However, the argument in their rebuttal is a bit defensive and beside the point. Corelet condensates of course rely on 'normal' self and self-other interactions, and it is amazing and wonderful that they recapitulate so many functions of endogenous Pc condensates. However, Corelet condensate formation also relies on light activation of oligomerization, in a manner that differs significantly from normal *in vivo* mechanisms, and should not be minimized as potentially affecting function.

Keep in mind that for a biologist, the claim that "We have shown that our light-activatable condensates recapitulate essentially every aspect of endogenous condensates as reported in literature." is only correct if the criteria is restricted to the ability to make a condensate that recruits the appropriate components and promotes enzymatic reactions *in vitro* or in cell culture (still impressive and exciting). Demonstrating that functionality is completely recapitulated has to include the ability to promote normal function in an organism, in this case the development of segment identity by Pc-mediated transcriptional silencing. The Corelet system could provide this ultimate measure of function, and it is worth testing this hypothesis directly (not for this paper). But it is important to acknowledge that the system is likely to fall short of complete functionality, given that forcing oligomerization through Corelets bypasses the normal temporal and spatial regulation of condensate formation and dissolution, as well as genome localization and gene expression, in a developing organism.

REVIEWER COMMENTS

Reviewer #1 (Remarks to the Author):

The authors properly addressed all raised issues of the original manuscript. Now, the revised work sufficiently support the model that the authors suggest. Therefore, I recommend publication of this work.

We thank the reviewer for the positive assessment of our revised manuscript.

Reviewer #2 (Remarks to the Author):

The authors have addressed each point raised in the initial review. I do not find some of their arguments convincing (particularly the insistence that these effects are dependent on PRC1 formation and hetero-oligomerization and the insistence that H2A119Ub is involved despite showing only correlations, not doing the experiments that would test this, and brushing aside results with RNF2 that seem at odds with this interpretation). Thus, I disagree with the data interpretation (or feel that it goes beyond what is shown), but nevertheless find the observations novel and interesting, and remain willing to be convinced by future experiments. There are still several points that should be addressed/corrected/clarified.

We thank the reviewer for emphasizing that our results are novel and interesting. Below we discuss the lingering disagreements with data interpretation.

1. The Western blot analysis in Figure S1B indicates that CBX2 is overexpressed by 20x, ≥ 4 x more than any of the other proteins. This should be taken into consideration when interpreting the condensates, particularly since CBX2 is the only protein that seems to compact chromatin. It is possible that given the low resolution of the analysis of chromatin compaction, higher expression levels of the other proteins could also allow chromatin compaction. The very high level of overexpression of CBX2 is also not consistent with its effects reflecting the activity of PRC1. Given the importance of the CBX2F12A mutant for the conclusions, overexpression of this protein should also be analyzed—i.e. to confirm that it's effects are due to the mutation and not lower expression.

The reviewer is correct, Cbx2 is an outlier in terms of overexpression levels. However, we note that this is an average taken over the bulk of cells. When we look at the microscopy data in more detail, we see that regardless of the expression level, Cbx2 tends to localize on the chromatin, even at low concentrations, whereas the other tested proteins show a different localization pattern, even at high expression. Although this analysis goes beyond the scope of our study, these observations on the absence of a correlation between chromatin localization and protein expression give us confidence in our interpretation. Thus, although we cannot formally rule out the possibility raised by the reviewer, our observations argue against it. We have added a sentence to revised manuscript to explicitly make the reader aware of the higher average overexpression level of CBX2:

"BMI1, PHC1 and RNF2 mCh-sspB fusion proteins were expressed <5 fold higher than the endogenous level, while CBX2 expression was significantly higher, roughly 20 fold"

As a more minor point, the western blots should be shown in standard format (not overlaid on the bar graph) with size markers, and the endogenous and overexpressed bands indicated.

We have included the uncropped western blots in the supporting data file, which also contains all other raw data. But for making our point in the manuscript, we feel the overlay on the bar graph conveys the message in the clearest possible way.

2. Figure 1N The E3 ligase activity is actually mediated by the Ring domain. RNF2 consists of a RING and RAWUL, very similar to BMI1, except without the small C-terminal IDR. This should be corrected. For clarification, Figure 2 of this review nicely shows the configuration of BMI-RING proteins vis-a-vis the E3 ligase activity.

<https://doi.org/10.1016/j.exphem.2016.12.006>

We thank the reviewer for this point – we have made the suggested correction.

3. Please clarify H2B-miRFP is H2B-miRFP670 in the text and figure legends (otherwise it appears that two red FPs are being used in the same experiment).

We have made the suggested clarification.

4. The construction of the phase diagrams in Figure 3 needs further explanation. If I understand the figure legend, each symbol is a single cell, so that many of the points seem to be represented by a very small number of cells. The number of cells analyzed at each point, and how many did versus did not form condensates should be provided as source data.

The reviewer is correct: each individual symbol is a single cell. We have provided all plotted points in the source data per the reviewer's request.

This is what is written in the methods section:

A standard imaging protocol was used on all cells to avoid variability. All activation protocols were 3 minutes with 2 second intervals. Only cells that were fully in the field of view were considered. The average GFP and mCh fluorescence intensity was determined using the first frame (before activation). Determination of whether or not cells were forming condensates was determined qualitatively by two independent observers, one of whom was blinded to experimental conditions. The assessments of the two observers were consistent in nearly all cases. The few cells on which observers disagreed were excluded from the results. FCS calibration curves were used to determine the mCh and GFP concentrations from the fluorescence intensities as described in Bracha et al (2018). Valence was measured as the ratio of sspB-fused protein to core.

Where does the GFP come from? Can images of GFP be provided to support the ratios? It is my understanding that in this experiment the iLid construct contains GFP (as in Bracha et al., 2018, Fig. 1), but in the other experiments (at least the GFP colocalization ones), it does not? Please state clearly which constructs were used in each experiment (or if possible include a schematic as in Bracha et al. Fig. 1). This is especially confusing in Fig. 3 because the colocalization with GFP-PcG proteins is intermixed with the phase diagrams.

For quantification of the phase diagrams, the iLID construct contains GFP. For colocalization experiments, we used an unlabeled iLID construct. We have clarified this in the manuscript, to avoid any ambiguity. We thank the reviewer for helping us make this as clear as possible.

Please move the legends off the graphs as it is not possible for the reader to know if they obscure data points.

We have made the suggested adjustment.

5. In figure 5, there is no "H" in the figure. It appears that the bottom part of F should be "G". More importantly, how does the model behave if there is no H2A119Ub?

We thank the reviewer for pointing out this mistake in panel labeling. We have corrected it in the revised manuscript. Regarding the question about the model behavior, we have added the suggested control to Figure 5F. As expected, compaction is not sustained when histones are not modified.

6. The description of image analysis (which is central to the conclusions of this paper) is not adequate to allow another group to reproduce it. The methods currently state:

Image analysis

All images were analysed using a combination of manual segmentation (ImageJ) and automated segmentation in Matlab.

We apologize for any confusion. The brief description in the previous methods subheading was potentially confusing, as additional details relevant to image analysis were included in the following methods subheading on phase diagram construction (with other imaging details also in the "Live Cell Imaging" subheading). To avoid confusion, we have combined these and included some additional details as follows:

“A standard imaging protocol was used on all cells to avoid variability. All activation protocols were 3 minutes with 2 second intervals. Only cells that were fully in the field of view were considered. Nuclei were manually segmented in ImageJ and the average GFP and mCh fluorescence intensity was determined using the first frame (before activation). Determination of whether or not cells were forming condensates was determined qualitatively by two independent observers, one of whom was blinded to experimental conditions. The assessments of the two observers were consistent in nearly all cases. The few cells on which observers disagreed were excluded from the results. FCS calibration curves were used to determine the mCh and GFP concentrations from the fluorescence intensities as described in Bracha et al (2018). Valence was measured as the ratio of sspB-fused protein to core. The code is available upon request, as stated in the data, code, and materials availability statement.”

7. For the description of the variance analysis :

For variance determination, the nucleus was segmented in every frame, with nucleoli excluded.

For each timepoint, the variance in pixel intensities was determined in each channel as:

Where n is the sample size (number of pixels), and i is the individual pixel index, with A^i the value in that pixel, and μ the mean. The variance over time was averaged over the amount of cells specified in the figure legend, and normalized.

What is meant by « the amount of cells specified in the figure legend »--i did not find information about the amount of cells in the figure legends. And how were the data normalized?

We apologize for any confusion. The caption of the figure indicates the number of replicates. We have clarified this in the revised manuscript. We have also clarified the normalization, as follows:

“The variance over time was normalized from 0 to 1 per trace, and we plot the result averaged over the number of cells specified in the figure caption.”

8. A note on nomenclature. Typically the human proteins are indicated in all caps (i.e. BMI1, CBX2); as written, the figures/text imply that the mouse proteins (Bmi1, Cbx2) were used. Unless this is the case, it would be good to stay with the standard nomenclature (as was done for RNF2 and PHC1).

We have made the suggested changes.

9. From the previous review :

1) The authors conclude that hetero-oligomerization drives formation of condensates but this is not clearly supported by their data. The literature suggests an alternative model, namely that both CBX2 and PHC1 can form condensates in cells that do not depend on PRC1 assembly (but can include PRC1 subunits). None of the experiments provided in this manuscript refute this interpretation.

Previous work on both Cbx2 (Ren and Kingston labs) and Phc2 (Koseki lab) has demonstrated that both of these proteins can form clusters/condensates in cells without interacting with other PRC1 subunits. For Cbx2, this was shown by analyzing condensates in cells lacking Ring1B/Ring1A (no PRC1 formed) as well as those lacking Bmi1/Mel18 (no canonical PRC1 formed). For Phc1 (Isono et al., Dev. Cell, 2013 doi:10.1016/j.devcel.2013.08.016.), clustering depends on the polymerization interfaces of the SAM, but not on the HD1 domain that mediates assembly into PRC1.

We thank the reviewer for these comments. We do not think that the data in the literature is in conflict with the data and physical picture we present here. Indeed, our findings show that oligomerization of individual PRC1 subunits can drive condensate assembly. Our findings that the synthetic PRC1 condensates can recruit other PRC1 components through hetero-oligomerization are not inconsistent with the papers cited above, however we realize that the language we used may be mis-interpreted and so we have altered the text to avoid this confusion.

I do not see that this response addresses the criticism. Particularly for CBX2, i do not find evidence supporting a requirement for PRC1 in condensate formation. It seems equally possible that most of what is observed is CBX2 alone, driven by the high level of overexpression (see note regarding sFig. X) It is unfortunate that the authors refuse to do simple IP experiments (which could be done with the reagents in hand) to at least begin to address what fraction of their CBX2 is in PRC1 versus on its own.

We apologize for the confusion, as we think this is a misunderstanding. Our data in Figure 3 shows that removal of hetero-oligomerization domains on BMI1 and PHC1 abolishes condensate formation. We therefore conclude that hetero-oligomerization is required for multicomponent PRC1 condensate

formation. As the reviewer correctly states, we do not show this result for CBX2 explicitly, and our results are therefore not in conflict with the literature. Our phrasing of the subheading “Hetero-oligomerization is required for condensate formation” may have contributed to this confusion. We have therefore modified this subheading, which now reads “Hetero-oligomerization contributes to condensate formation”.

Minor points:

1. Fig. S2B PHC1 Δ OD-GFP is labelled as “ Δ OD” in legend but “ Δ SAM” on image

We have corrected this in the manuscript.

2. In the source data for Figure S3G, the blot is labelled “H2AK119Ub”, but the title is “Quantification Western Blot H3K27me3”

We have corrected this in the source data.

Reviewer #3 (Remarks to the Author):

This reviewer is satisfied with the majority of the responses and revisions, and fully supports publication of this manuscript.

We thank the reviewer for this positive assessment and recommendation for publication.

However, it is worth clarify points concerning interpretations (not the data or primary conclusions, which are sound). My hope is that the authors will step back and consider that there are in fact limitations to the system and findings, as well as alternative explanations for the results, that are worth mentioning in the Discussion. Consider this an academic discussion, not a critique of the content or validity of the paper, nor a requirement for publication.

We thank the reviewer for thoughtfully outlining these potential points for inclusion in the discussion. We are particularly keen to broaden the discussion of both the power and potential limitations of the synthetic Corelet approach. Toward this end, we have now included the following language in the discussion section:

“We note that this approach uses light-dependent optogenetic proteins to oligomerize phase separation-prone proteins, and may exhibit differences from the biologically-regulated oligomerization important for the formation and function of native condensates.”

It is exciting and appropriate to claim that the condensate per se is not required to maintain compaction in this system. However, the authors should acknowledge that it could just as easily require both the marks and the reader proteins, even if they are not forming a condensate (eg compaction via local crosslinks mediated by the readers). The emphasis on the sufficiency of histone marks for chromatin compaction is certainly strengthened by the modeling, but it is not validated experimentally. That requires the demonstration that compaction occurs in the presence of the marks after eliminating or interfering with the presence or function of the reader proteins.

We agree with the reviewer that we cannot confirm the direct link between the histone marks and compaction. We have emphasized this in the discussion per the reviewer’s suggestion, by including the following sentence:

“Future experiments should be aimed to confirm the direct link between repressive histone marks and compaction.”

In addition, the authors’ vigorous defense of the Corelet system is understandable and appropriate; this and other optosystems developed in the Brangwynne lab and elsewhere are powerful. However, the argument in their rebuttal is a bit defensive and beside the point. Corelet condensates of course rely on ‘normal’ self and self-other interactions, and it is amazing and wonderful that they recapitulate so many functions of endogenous Pc condensates. However, Corelet condensate formation also relies on light activation of oligomerization, in a manner that differs significantly from normal in vivo mechanisms, and should not be minimized as potentially affecting function.

These are all fair points and we agree (see the newly-included Discussion sentence above).

Keep in mind that for a biologist, the claim that “We have shown that our light-activatable condensates recapitulate essentially every aspect of endogenous condensates as reported in literature.” is only correct if the criteria is restricted to the ability to make a condensate that recruits the appropriate components and promotes enzymatic reactions in vitro or in cell culture (still impressive and exciting). Demonstrating that functionality is completely recapitulated has to include the ability to promote normal function in an organism, in this case the development of segment identity by Pc-mediated transcriptional silencing. The Corelet system could provide this ultimate measure of function, and it is worth testing this hypothesis directly (not for this paper). But it is important to acknowledge that the system is likely to fall short of complete functionality, given that forcing oligomerization through Corelets bypasses the normal temporal and spatial regulation of condensate formation and dissolution, as well as genome localization and gene expression, in a developing organism.

We thank the reviewer for this feedback. We acknowledge that, indeed, we have not shown function in a developing organism, which is definitely a goal we have discussed and aspire to achieve in our future work. We appreciate the reviewer pointing out that for a biologist, this means we cannot claim we recapitulated functionality completely, and we will keep this in mind for future experiments and projects.

REVIEWERS' COMMENTS

Reviewer #2 (Remarks to the Author):

The authors have revised the text to address the concerns raised. I look forward to seeing this nice work published, although I have one additional point of confusion, and a small point on Fig. 1

Fig. 1N--the schematic for RING is inverted--like Bmi1, the order of domains is Ring-Rawul (and the E3 ligase domain should be removed).

I remain confused as to why the authors refer to the PHC1 SAM as a hetero-oligomerization domain, as it forms homo-oligomers (although at least in *Drosophila* can form hetero-oligomers with another SAM-containing PcG protein, SCM). The SAM does not mediate assembly into PRC1, the HD1 does. Thus, while this experiment shows SAM-dependent phase separation, it does not show that this involves PRC1. Indeed, PHC1 without the SAM should still assemble into PRC1 (based on published data). Thus, in the experiments where the SAM is deleted either PRC1 requires PHC1 SAM homotypic interactions for phase separation, or PHC1 SAM-dependent phase separation does not involve PRC1. Given that this is not the main point of the paper, it would be simple to modify the text. However, I leave it to the authors to decide how to interpret their results, as I may be misunderstanding the argument.

REVIEWERS' COMMENTS

Reviewer #2 (Remarks to the Author):

The authors have revised the text to address the concerns raised. I look forward to seeing this nice work published, although I have one additional point of confusion, and a small point on Fig. 1

We thank the reviewer for their enthusiasm, and particularly for their help in improving this manuscript.

Fig. 1N--the schematic for RING is inverted--like Bmi1, the order of domains is Ring-Rawul (and the E3 ligase domain should be removed).

Thank you - we have made the suggested change.

I remain confused as to why the authors refer to the PHC1 SAM as a hetero-oligomerization domain, as it forms homo-oligomers (although at least in *Drosophila* can form hetero-oligomers with another SAM-containing PcG protein, SCM). The SAM does not mediate assembly into PRC1, the HD1 does. Thus, while this experiment shows SAM-dependent phase separation, it does not show that this involves PRC1. Indeed, PHC1 without the SAM should still assemble into PRC1 (based on published data). Thus, in the experiments where the SAM is deleted either PRC1 requires PHC1 SAM homotypic interactions for phase separation, or PHC1 SAM-dependent phase separation does not involve PRC1. Given that this is not the main point of the paper, it would be simple to modify the text. However, I leave it to the authors to decide how to interpret their results, as I may be misunderstanding the argument.

Deletion of the SAM domain results in reduced recruitment to other PRC1 condensates formed by BMI1, PHC1WT, or RNF2. We interpret this as PRC1 proteins failing to oligomerize with PHC1 protein. We are aware this has not been shown in literature, but we choose to stay with this interpretation.